# DATA-DRIVEN DISCOVERY OF PDES VIA THE ADJOINT METHOD

## ABSTRACT

In this work, we present an adjoint-based method for discovering the underlying governing partial differential equations (PDEs) given data. The idea is to consider a parameterized PDE in a general form and formulate a PDE-constrained optimization problem aimed at minimizing the error of the PDE solution from data. Using variational calculus, we obtain an evolution equation for the Lagrange multipliers (adjoint equations) allowing us to compute the gradient of the objective function with respect to the parameters of PDEs given data in a straightforward manner. In particular, we consider a family of parameterized PDEs encompassing linear, nonlinear, and spatial derivative candidate terms, and elegantly derive the corresponding adjoint equations. We show the efficacy of the proposed approach in identifying the form of the PDE up to machine accuracy, enabling the accurate discovery of PDEs from data. We also compare its performance with the famous PDE Functional Identification of Nonlinear Dynamics method known as PDE-FIND Rudy et al. (2017), on both smooth and noisy data sets. Even though the proposed adjoint method relies on forward/backward solvers, it outperforms PDE-FIND for large data sets thanks to the analytic expressions for gradients of the cost function with respect to each PDE parameter.

## 1 INTRODUCTION

A large portion of data-driven modelling of physical processes in literature is dedicated to deploying Neural Networks to obtain fast prediction given the training data set. The data-driven estimation methods include Physics-Informed Neural Networks Raissi et al. (2019), Pseudo-Hamiltonian neural networks Eidnes and Lye (2024), structure preserving Matsubara et al. (2020); Sawant et al. (2023), and reduced order modelling Duan and Hesthaven (2024). These methods often provide efficient and somewhat "accurate" predictions when tested as an interpolation method in the space of input or boundary parameters. Such fast estimators are beneficial when many predictions of a dynamic system is needed, for example in the shape optimization task in fluid dynamics.

However, the data-driven estimators often fail to provide accurate solution to the dynamical system when tested outside the training space, i.e. for extrapolation. Furthermore, given the regression-based nature of these predictors, often they do not offer any error estimator in prediction. Since we already have access to an arsenal of numerical methods in solving traditional governing equations, it is attractive to learn the underlying governing equation given data instead. Once the governing equation is found, one can use the standard and efficient numerical methods for prediction. This way we guarantee the consistency with observed data, estimator for the numerical approximation, and interoperability. Hence, learning the underlying physics given data has motivated a new branch in the scientific machine learning for discovering the mathematical expression as the governing equation given data.

The wide literature of data-driven discovery of dynamical systems includes equation-free modeling Kevrekidis et al. (2003), artificial neural networks González-García et al. (1998), nonlinear regression Voss et al. (1999), empirical dynamic modeling Sugihara et al. (2012); Ye et al. (2015), modeling emergent behavior Roberts (2014), automated inference of dynamics Schmidt et al. (2011); Daniels and Nemenman (2015a;b), normal form identification in climate Majda et al. (2009), nonlinear Laplacian spectral analysis Giannakis and Majda (2012) and Koopman analysis Mezić (2013) among others. There has been a significant advancement in this field by combining symbolic regression

with the evolutionary algorithms Bongard and Lipson (2007); Schmidt and Lipson (2009); Tohme et al. (2022), which enable the direct extraction of nonlinear dynamical system information from data. Furthermore, the concept of sparsity Tibshirani (1996) has recently been employed to efficiently and robustly deduce the underlying principles of dynamical systems Brunton et al. (2016); Mangan et al. (2016).

**Related work.**   Next, we review several relevant works that have shaped the current landscape of discovering PDEs from data:

*PDE-FIND* Rudy et al. (2017).  This method has been developed to discover underlying partial differential equation by minimizing the $L_2^2$-norm point-wise error of the parameterized forward model from the data. Estimating all the possible derivatives using Finite Difference, PDE-FIND constructs a dictionary of possible terms and finds the underlying PDE by performing a sparse search using ridge regression problem with hard thresholding, also known as STRidge optimization method. Several further developments in the literature has been carried out based on this idea, namely Champion et al. (2019); Kaheman et al. (2020). In these methods, as the size (or dimension) of the data set increases, the PDE discovery optimization problem based on point-wise error becomes extremely expensive, forcing the user to arbitrarily reduce the size of data by resampling, or compress the data using proper orthogonal decomposition. Needles to say, in case of non-linear dynamics, such truncation of data can introduce bias in prediction leading to finding a wrong PDE.

*PDE-Net* Long et al. (2018; 2019). In this method, the PDE is learned from data using convolution kernels rather than brute-force use of Finite Differences, and apply neural networks to approximate nonlinear responses. Similar to PDE-FIND, the loss function of PDE-Net is the point-wise error from data which leads to a regression task that does not scale well with the size of the data set.

*Hidden Physics Models* Raissi and Karniadakis (2018). This method assumes that the relevant terms of the governing PDE are already identified and finds its unknown parameters using Gaussian process regression (GPR). While GPR is an accurate interpolator which offers an estimate for the uncertainty in prediction, its training scales poorly with the size of the training data set as it requires inversion of the covariance matrix.

*PINN-SR* Chen et al. (2021). One of the issues with the PDE-FIND is the use of Finite Difference in estimating the derivatives. The idea of PINN-SR is to extend PDE-FIND's optimization problem to also find a PINN fit to the data in order to find smooth estimates for spatial derivatives. In particular, the training of PINN-SR combines the search for weights/biases of PINN approximation of the PDE with the sparse search in the space of possible terms to find the coefficients of the PDE given data. However, similar to PDE-FIND, the point-wise error from data is used as the loss for the regression task which does not scale well with the size of the data set.

**Contributions.**   In this paper, we introduce a novel approach for discovering PDEs from data based on the well-known adjoint method, i.e. PDE-constrained optimization method. The idea is to formulate the objective (or cost) functional such that the estimate function $f$ minimizes the $L_2^2$-norm error from the data points $f^*$ with the constraint that $f$ is the solution to a parameterized PDE using the method of Lagrange multipliers. Here, we consider a parameterized PDE in a general form and the task is to find all the parameters including irrelevant ones. By finding the variational extremum of the cost functional with respect to the function $f$, we obtain a backward-in-time evolution equation for the Lagrange multipliers (adjoint equations). Next, we solve the forward parameterized PDE as well as the adjoint equations numerically. Having found estimates of the Lagrange multipliers and solution to the forward model $f$, we can numerically compute the gradient of the objective function with respect to the parameters of PDEs given data in a straightforward manner. In particular, for a family of parameterized and nonlinear PDEs, we show how the corresponding adjoint equations can be elegantly derived. We note that the adjoint method has been successfully used before as an efficient method for uncertainty quantification Flath et al. (2011), shape optimization and sensitivity analysis method in fluid mechanics Jameson (2003); Caflisch et al. (2021) and plasma physics Antonsen et al. (2019); Geraldini et al. (2021). Unlike the usual use of PDE-constrained adjoint optimization where the governing equation is known, in this paper we are interested in finding the form along with the coefficients of the PDE given data.

The remainder of the paper is organized as follows. First in Section 2, we introduce and derive the proposed adjoint-based method of finding the underlying system of PDEs given data. Next in Section

3, we present our results on a wide variety of PDEs and compare the solution with the celebrated PDE-FIND in terms of error and computational/training time. In Section 7, we discuss the limitations for the current version of our approach and provide concluding remarks in Section 8.

## 2  ADJOINT METHOD FOR FINDING PDEs

In this section, we introduce the problem and derive the proposed adjoint method for finding governing equations given data.

**Problem setup.**  Assume we are given a data set on a spatial/temporal grid $\mathcal{G} = \bigcup_{j=0}^{N_t} \mathcal{G}^{(j)}$ with $\mathcal{G}^{(j)} = \{(\boldsymbol{x}^{(k)}, t^{(j)}) \mid k = 1, ..., N_{\boldsymbol{x}}\}$ for the vector of functions $\boldsymbol{f}^*$ where $k$ is the spatial index and $j$ the time index with $t^{(N_t)} = T$ being the final time. Here, $\boldsymbol{x}^{(k)} \in \Omega \subset \mathbb{R}^n$ is coordinates inside the solution domain $\Omega$, $t^{(j)}$ denotes the $j$-th time that data is available, and output is a discrete map $\boldsymbol{f}^* : \mathcal{G} \to \mathbb{R}^N$. The goal is to find the governing equations that accurately estimates $\boldsymbol{f}^*$ at all points on $\mathcal{G}$. In order to achieve this goal, we formulate the problem using the method of Lagrange multipliers.

**Adjoint method.**  For simplicity, let us first consider only the time interval $t \in [t^{(j)}, t^{(j+1)}]$. Consider a general a forward model $\mathcal{L}[\cdot]$ that evolves an $N$-dimensional vector of continuous functions $\boldsymbol{f}(\boldsymbol{x}, t = t^{(j)})$ in $t \in (t^{(j)}, t^{(j+1)}]$ and $\boldsymbol{x} \in \Omega$ where the $i$-th PDE is given by

$$\mathcal{L}_i[\boldsymbol{f}] := \partial_t f_i + \sum_{\boldsymbol{d}, \boldsymbol{p}} \alpha_{i,\boldsymbol{d},\boldsymbol{p}} \nabla_{\boldsymbol{x}}^{(\boldsymbol{d})}[\boldsymbol{f}^{\boldsymbol{p}}] = 0 \tag{1}$$

for $i = 1, \ldots, N$, resulting in a system of N-PDEs, i.e. the $i$-th PDE $\mathcal{L}_i$ predicts $f_i$. Here, $\boldsymbol{x} = [x_1, x_2, \ldots, x_n]$ is an n-dimensional (spatial) input vector, and $\boldsymbol{f} = [f_1, f_2, \ldots, f_N]$ is an N-dimensional vector of functions. We use the shorthand $f_i = f_i(\boldsymbol{x}, t)$ and $\boldsymbol{f} = \boldsymbol{f}(\boldsymbol{x}, t)$. Furthermore, $\boldsymbol{p} = [p_1, \ldots, p_N]$ and $\boldsymbol{d} = [d_1, d_2, \ldots, d_n]$ are non-negative index vectors such that $\boldsymbol{f}^{\boldsymbol{p}} = f_1^{p_1} f_2^{p_2} \cdots f_N^{p_N}$ and

$$\nabla_{\boldsymbol{x}}^{(\boldsymbol{d})} = \nabla_{x_1}^{(d_1)} \nabla_{x_2}^{(d_2)} \cdots \nabla_{x_n}^{(d_n)} \; , \tag{2}$$

where $\nabla_{x_i}^{(d_i)}$ for $i = 1, ..., n$ indicates $d_i$-th derivative in $x_i$ dimension, and $\partial_t f_i$ denotes the time derivative of the $i$-th function. We denote the vector of unknown parameters by $\boldsymbol{\alpha} = [\alpha_{i,\boldsymbol{d},\boldsymbol{p}}]_{(i,\boldsymbol{d},\boldsymbol{p}) \in \mathcal{D}}$, where $\mathcal{D}$ represents the domain of all valid combinations of $i$, $\boldsymbol{d}$, and $\boldsymbol{p}$.

Having written the forward model equation 1 as general as possible, the goal is to find the parameters $\boldsymbol{\alpha}$ such that $\boldsymbol{f}$ approximates the data points of $\boldsymbol{f}^*$ at $t = t^{(j+1)}$ given the solution $\boldsymbol{f} = \boldsymbol{f}^*$ at $t = t^{(j)}$. To this end, we formulate a semi-discrete objective (or cost) functional that minimizes the $L_2^2$-norm error between what the model predicts and the data $\boldsymbol{f}^*$ on $\mathcal{G}^{(j+1)}$, with the constraint that $\boldsymbol{f}$ solves the forward model in Eq. (1), i.e.

$$\mathcal{C} = \sum_{i=1}^{N} \Bigg( \sum_k \Big( f_i^*\big(\boldsymbol{x}^{(k)}, t^{(j+1)}\big) - f_i\big(\boldsymbol{x}^{(k)}, t^{(j+1)}\big) \Big)^2$$

$$+ \frac{1}{\Delta \boldsymbol{x} \Delta t} \int \lambda_i(\boldsymbol{x}, t) \mathcal{L}_i[\boldsymbol{f}(\boldsymbol{x}, t)] d\boldsymbol{x} dt \Bigg) + \epsilon_0 ||\boldsymbol{\alpha}||_2^2 \; , \tag{3}$$

where $\Delta \boldsymbol{x}$ and $\Delta t$ denotes grid spacing in $\Omega$ and step size in $t$, respectively, $||.||_2$ denotes $L_2$-norm, and $\epsilon_0$ is the regularization factor. We note that PDE discovery task is ill-posed since the underlying PDE is not unique and the regularization term helps us find the PDE with the least possible coefficients.

Clearly, given estimates of $\boldsymbol{f}$ and Lagrange multipliers $\boldsymbol{\lambda} = (\lambda_1, \lambda_2, \ldots, \lambda_N)$, the gradient of the cost function with respect to model parameters can be simply computed via

$$\frac{\partial \mathcal{C}}{\partial \alpha_{i,\boldsymbol{d},\boldsymbol{p}}} = (-1)^{|\boldsymbol{d}|} \frac{1}{\Delta \boldsymbol{x} \Delta t} \int \boldsymbol{f}^{\boldsymbol{p}} \nabla_{\boldsymbol{x}}^{(\boldsymbol{d})}[\lambda_i] d\boldsymbol{x} dt + 2\epsilon_0 \alpha_{i,\boldsymbol{d},\boldsymbol{p}} \tag{4}$$

where $i = 1, ..., N$ and $|\boldsymbol{d}| = d_1 + ... + d_n$, where $|.|$ denotes $L_1$-norm. Here, we used integration by parts and imposed the condition that $\boldsymbol{\lambda} \to \boldsymbol{0}$ on the boundaries of $\Omega$ at all time $t \in [0, T]$. The

analytical expression equation 4 can be used for finding the parameters of PDE using in the gradient descent method with update rule

$$\alpha_{i,\boldsymbol{d},\boldsymbol{p}} \leftarrow \alpha_{i,\boldsymbol{d},\boldsymbol{p}} - \eta \frac{\partial \mathcal{C}}{\partial \alpha_{i,\boldsymbol{d},\boldsymbol{p}}} \tag{5}$$

for $i = 1, \ldots, N$, where $\eta = \beta \min(\Delta \boldsymbol{x})^{|\boldsymbol{d}| - d_{\max}}$ is the learning rate which includes a free parameter $\beta$ and scaling coefficient for each term of the PDE, and $d_{\max} = \max(|\boldsymbol{d}|)$ for all considered $\boldsymbol{d}$. Let us also define $p_{\max} = \max(|\boldsymbol{p}|)$ as the highest order in the forward PDE model. We note that since the terms of the PDEs may have different scaling, the step size for the corresponding coefficient must be adjusted accordingly. This is due the fact that the gradient of the cost function is most sensitive to the highest order terms of the PDE. In Appendix E, we give a justification for our choice of the learning rate $\eta$.

However, before we can use Eq. (4) and (5), we need to find $\boldsymbol{\lambda}$, hence the adjoint equation. This can be achieved by finding the functional extremum of the cost functional $\mathcal{C}$ with respect to $\boldsymbol{f}$. First, we note that the semi-descrete total variation of $\mathcal{C}$ can be derived as

$$\begin{aligned}
\delta \mathcal{C} = \sum_{i=1}^{N} \Bigg( &-\sum_{k} 2\Big(f_i^*\big(\boldsymbol{x}^{(k)}, t^{(j+1)}\big) - f_i\big(\boldsymbol{x}^{(k)}, t^{(j+1)}\big)\Big) \delta f_{i,\boldsymbol{x}^{(k)}, t^{(j+1)}} \\
&+ \frac{1}{\Delta \boldsymbol{x} \Delta t} \int \Bigg( -\frac{\partial \lambda_i}{\partial t} + \sum_{\boldsymbol{d},\boldsymbol{p}} (-1)^{|\boldsymbol{d}|} \alpha_{i,\boldsymbol{d},\boldsymbol{p}} \nabla_{f_i}[\boldsymbol{f}^{\boldsymbol{p}}] \nabla_{\boldsymbol{x}}^{(\boldsymbol{d})}[\lambda_i] \Bigg) \delta f_i d\boldsymbol{x} dt \\
&+ \sum_{k} \lambda_i\big(\boldsymbol{x}^{(k)}, t^{(j+1)}\big) \delta f_{i,\boldsymbol{x}^{(k)}, t^{(j+1)}} \Bigg)
\end{aligned} \tag{6}$$

where $\delta f_i$ denotes variation with respect to $f_i(\boldsymbol{x}, t)$, and $\delta f_{i,\boldsymbol{x}^{(k)}, t^{(j+1)}}$ variation with respect to $f_i(\boldsymbol{x} = \boldsymbol{x}^{(k)}, t = t^{(j+1)})$. In this derivation, we descretized the last integral resulting from integration by parts in time using the same mesh as the one of data $\mathcal{G}^{(j+1)}$. Here again, we used integration by parts and imposed the condition that $\boldsymbol{\lambda} \to \boldsymbol{0}$ on the boundaries of $\Omega$ at all time $t \in [t^{(j)}, t^{(j+1)}]$ for $i = 1, ..., N$. Note that $f_i(\boldsymbol{x}, t)$ is the output of $i$-th PDE.

Next, we find the optimums of $\mathcal{C}$ (and hence the adjoint equations) by taking the variational derivatives with respect to $f_i$ and $f_{i,\boldsymbol{x}^{(k)}, t^{(j+1)}}$, i.e.

$$\frac{\delta \mathcal{C}}{\delta f_i} = 0 \implies \frac{\partial \lambda_i}{\partial t} = \sum_{\boldsymbol{d},\boldsymbol{p}} (-1)^{|\boldsymbol{d}|} \alpha_{i,\boldsymbol{d},\boldsymbol{p}} \nabla_{f_i}[\boldsymbol{f}^{\boldsymbol{p}}] \nabla_{\boldsymbol{x}}^{(\boldsymbol{d})}[\lambda_i] \tag{7}$$

and

$$\frac{\delta \mathcal{C}}{\delta f_{i,\boldsymbol{x}^{(k)}, t^{(j+1)}}} = 0 \implies \lambda_i(\boldsymbol{x}^{(k)}, t^{(j+1)}) = 2(f_i^*(\boldsymbol{x}^{(k)}, t^{(j+1)}) - f_i(\boldsymbol{x}^{(k)}, t^{(j+1)})) \tag{8}$$

for $i = 1, ..., N$ and $j = 0, ..., N_t - 1$. We note that the adjoint equation equation 7 for the system of PDEs is backward in time with the final condition at the time $t = t^{(j+1)}$ given by Eq. (8). In order to make the notation clear, we present examples for deriving the adjoint equations in Appendix F. The adjoint equation is in the continuous form, while the final condition is on the discrete points, i.e. on the grid $\mathcal{G}^{(j+1)}$. In order to obtain the Lagrange multipliers in $t \in [t^{(j+1)}, t^{(j)}]$, a numerical method appropriate for the forward equation 1 and adjoint equation equation 7 should be deployed. Furthermore, the adjoint equation should have the same or coarser spatial discretization as $\mathcal{G}^{(j+1)}$ to enforce the final condition equation 8.

**Training with smooth data set.** The training procedure follows the standard gradient descent method. We start by taking an initial guess for parameters $\boldsymbol{\alpha}$, e.g. here we take $\boldsymbol{\alpha} = 0$ initially. For each time interval $t \in [t^{(j)}, t^{(j+1)}]$, first we solve the forward model equation 1 numerically to estimate $\boldsymbol{f}(\boldsymbol{x}^{(k)}, t^{(j+1)})$ given the initial condition

$$\boldsymbol{f}(\boldsymbol{x}^{(k)}, t^{(j)}) = \boldsymbol{f}^*(\boldsymbol{x}^{(k)}, t^{(j)}) . \tag{9}$$

Then, the adjoint Eq. equation 7 is solved backwards in time with the final time condition equation 8. Finally, the estimate for parameters of the model is updated using Eq. equation 5. We repeat this for

all time intervals $j = 0, ..., N_t - 1$ until convergence. In order to improve the search for coefficients and enforce the PDE identification, we also deploy thresholding, i.e. set $\alpha_{i,\boldsymbol{d},\boldsymbol{p}} = 0$ if $|\alpha_{i,\boldsymbol{d},\boldsymbol{p}}| < \sigma$ where $\sigma$ is a user-defined threshold, during and at the end of training, respectively. In Algorithm 1, we present a pseudocode for finding the parameters of the system of PDEs using the Adjoint method (a flowchart is also shown in Fig. 5 of Appendix A). For the introduced hyperparameters, we note that $\beta$ in the learning rate needs to be small enough to avoid unstable intermediate guessed PDEs, $\epsilon_0$ must be large enough to ensure uniqueness in cases where more than one solution may exist, and the thresholding should be applied only when solution to the optimization is not improving anymore up to a user-defined tolerance $\gamma_{\text{thr}}$. For suggested default values, please see the description of the algorithm. For experimental investigation on impact of these parameters for a few examples, see Appendix D.

We note that the type of guessed PDE may change during the training, which adds numerical complexity to the optimization and motivates the use of an appropriate solver for each type of guessed PDE, e.g. Finite Volume method for hyperbolic and Finite Element method for Elliptic PDEs. For simplicity, in this work we use the second order Finite Difference method across the board to estimate the spatial and Euler for the time derivative with small enough time step sizes in solving the forward/backward equations to avoid blowups due to possible instabilities. We note that the adjoint method is most effective when there is some prior knowledge of the underlying PDE type, and a suitable numerical method is deployed.

---

**Algorithm 1:** Finding system of PDEs using Adjoint method. Default threshold $\sigma = 10^{-3}$ applied after $N_{\text{thr}} = 100$ iterations, with tolerances $\gamma = 10^{-9}$ and $\gamma_{\text{thr}} = 10^{-6}$, and regularization factor $\epsilon_0 = 10^{-12}$.

---

**Input:** data $\boldsymbol{f}^*$, learning rate $\eta$, tolerance $\gamma$, threshold $\sigma$ applied after $N_{\text{thr}}$, and $\epsilon_0$
Initialize the parameters $\boldsymbol{\alpha} = \boldsymbol{0}$;
**repeat**
    **for** $j = 0, \dots, N_t - 1$ **do**
        Estimate $\boldsymbol{f}$ in $t \in \left(t^{(j)}, t^{(j+1)}\right]$ by solving forward model (1) given initial condition (9);
        Find $\boldsymbol{\lambda}$ in $t \in \left[t^{(j)}, t^{(j+1)}\right)$ by solving the adjoint equation in Eq. (7);
        Compute the gradient using Eq. (4);
        Update parameters $\boldsymbol{\alpha}$ using Eq. (5);
    **end**
    **if** Epochs $> N_{\text{thr}}$ *or convergence in* $\boldsymbol{\alpha}$ *with* $\gamma_{\text{thr}}$ **then**
        Thresholding: set $\alpha_i = 0$ for all $i$ that $|\alpha_i| < \sigma$;
    **end**
**until** *Convergence in* $\boldsymbol{\alpha}$ *with tolerance* $\gamma$;
Thresholding: set $\alpha_i = 0$ for all $i$ that $|\alpha_i| < \sigma$;
**Output:** $\boldsymbol{\alpha}$

---

**Training with noisy data set.** Often the data set comes with some noise. There are several pre-processing steps that can be done to reduce the noise at the expense of introducing bias, for example removing high frequencies using Fast Fourier Transform or removing small singular values from data set using Singular Value Decomposition. However, we can also reduce the sensitivity of the training algorithm to the noise by averaging the gradients before updating the parameters. Assuming that the noise is martingale, the Monte Carlo averaging gives us the unbiased estimator for the expected value of the gradient over all the data set. We adapt the training procedure by averaging gradients over all available data points and then updating the parameters (see Algorithm 2 and the flowchart in Fig. 5 of Appendix A for more details). Clearly, this will make the algorithm more robust at higher cost since the update happens only after seeing all the data.

## 3 RESULTS

We demonstrate the validity of our proposed adjoint-based method in discovering PDEs given measurements on a spatial-temporal grid. As a show case, next we show the results of the PDE discovery task applied to the heat equation. We also consider data collected from a variety of problems, including the Burgers', Kuramoto Sivashinsky, Random Walk, and Reaction Diffusion equation,

---

**Algorithm 2:** Finding a system of PDEs using the Adjoint method with averaging for the computation of gradients over data set. Default threshold $\sigma = 10^{-3}$ applied after $N_{\text{thr}} = 100$ iterations, with tolerances $\gamma = 10^{-9}$ and $\gamma_{\text{thr}} = 10^{-6}$, and regularization factor $\epsilon_0 = 10^{-12}$.

---

**Input:** data $\boldsymbol{f}^*$, learning rate $\eta$, tolerance $\gamma$, threshold $\sigma$ applied after $N_{\text{thr}}$, and $\epsilon_0$
Initialize the parameters $\boldsymbol{\alpha} = \boldsymbol{0}$;
**repeat**
    **for** $j = 0, \ldots, N_t - 1$ **do**
        Estimate $\boldsymbol{f}$ in $t \in \left( t^{(j)}, t^{(j+1)} \right]$ by solving forward model (1) given initial condition (9);
        Find $\boldsymbol{\lambda}$ in $t \in \left[ t^{(j)}, t^{(j+1)} \right)$ by solving the adjoint equation in Eq. (7);
        Compute the gradient $\boldsymbol{g}^{(j)} = \partial C^{(j)} / \partial \boldsymbol{\alpha}$ using Eq. (4);
    **end**
    Average the gradient $\mathbb{E}[\partial C / \partial \boldsymbol{\alpha}] = \sum_j \boldsymbol{g}^{(j)} / N_t$;
    Update parameters $\boldsymbol{\alpha}$ with Eq. (5) using $\mathbb{E}[\partial C / \partial \boldsymbol{\alpha}]$;
    **if** Epochs $> N_{\text{thr}}$ *or convergence in* $\boldsymbol{\alpha}$ *with* $\gamma_{\text{thr}}$ **then**
        Thresholding: set $\alpha_i = 0$ for all $i$ that $|\alpha_i| < \sigma$;
    **end**
**until** *Convergence in* $\boldsymbol{\alpha}$ *with tolerance* $\gamma$;
Thresholding: set $\alpha_i = 0$ for all $i$ that $|\alpha_i| < \sigma$;
**Output:** $\boldsymbol{\alpha}$

---

detailed in Appendix B. We have compared our approach to PDE-FIND in terms of error and time to convergence. All the results are obtained using a single core-thread of a 2.3 GHz Quad-Core Intel Core i7 CPU. In this paper, we report the execution time $\tau$ obtained with averaging over 10 independent runs and we use error bars to show the standard deviation of the expected time, i.e. $d_{\text{error-bar}} = \sqrt{\mathbb{E}[(\tau - \mathbb{E}[\tau])^2]}$.

### 3.1 HEAT EQUATION

As a first example, let us consider measured data collected from the solution to the heat equation, i.e.

$$\frac{\partial f}{\partial t} + D \frac{\partial^2 f}{\partial x^2} = 0, \tag{10}$$

with $D = -1$. The data is constructed using the Finite Difference method with initial condition $f(x, 0) = 5 \sin(2\pi x)x(x - L)$ and a mesh with $N_x = 100$ nodes in $x$ covering the domain $\Omega = [0, L]$ with $L = 1$ and $N_t = 100$ steps in $t$ with final time $T = N_t \Delta t$ where $\Delta t = 0.05 \Delta x^2 / (1 + |D|)$ is the step size and $\Delta x = L/N_x$ is the mesh size in $x$.

We consider a system consisting of a single PDE (i.e. $N = 1$, $\boldsymbol{f} = f$, and $\boldsymbol{p} = p$) with one-dimensional input, i.e. $n = 1$ and $\boldsymbol{x} = x \subset \mathbb{R}$, and $\boldsymbol{d} = d \in \mathbb{N}$. In order to construct a general forward model, here we consider derivatives and polynomials with indices $d, p \in \{1, 2, 3\}$ as the initial guess for the forward model. This leads to 9 terms with unknown coefficients $\boldsymbol{\alpha}$ that we find using the proposed adjoint method (an illustrative derivation of the candidate terms can be found in Appendix F.1). While we expect to recover the coefficient that corresponds to $D$, we expect all the other coefficients (denoted by $\boldsymbol{\alpha}^*$) to become negligible. That is what we indeed observe in Fig. 1 where the error of the coefficient for each term is plotted against the number of epochs.

Next, we compare the solution obtained via the adjoint method against PDE-FIND with STRidge optimization method. Here, we test both methods in recovering the heat equation given data on the grid with discretization $(N_t, N_x) \in \{(100, 100), (500, 100), (1000, 100), (1000, 1000)\}$. As shown in Fig. 1, the proposed adjoint method provides more accurate results across all data sizes. We also point out that as the size of the data set increases, PDE-FIND with STRidge regression method becomes more expensive, e.g. one order of magnitude more expensive than the adjoint method for the data on a grid size $(N_t, N_x) = (1000, 1000)$.

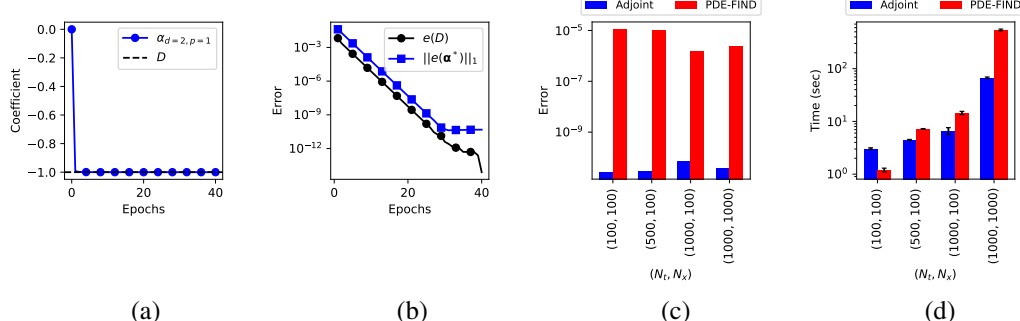

Figure 1: The estimated coefficient corresponding to $D$ in heat equation (a) and the $L_1$-norm error of all considered coefficients (b) using the proposed Adjoint method with $N_t = 100$ and $N_x = 100$. Also, we show $L_1$-norm error of the estimated coefficients (c) and the execution time (d) using the proposed Adjoint method (blue) and PDE-FIND method (red), given data on a grid with $N_t \in \{100, 500, 1000\}$ steps in $t$, and $N_x \in \{100, 1000\}$ nodes in $x$.

Table 1: Recovering the heat equation given sparse data set in time. Here we rounded the coefficients up to three decimals.

| $\%N_t$ | Method | Recovered PDE |
|---|---|---|
| 100 | Adjoint | $f_t - f_{xx} = 0$ |
| | PDE-FIND | $f_t - f_{xx} = 0$ |
| 50 | Adjoint | $f_t - f_{xx} = 0$ |
| | PDE-FIND | $f_t - 0.999f_{xx} + 0.177f - 0.261f^3 - 0.089ff_x - 0.011f^3f_x - 0.003f^2f_{xx} - 0.001ff_{xxx} = 0$ |
| 25 | Adjoint | $f_t - f_{xx} = 0$ |
| | PDE-FIND | $f_t - 0.999f_{xx} + 0.532f - 0.778f^3 - 0.268ff_x - 0.035f^3f_x - 0.010f^2f_{xx} - 0.003ff_{xxx} = 0$ |
| 12.5 | Adjoint | $f_t - f_{xx} = 0$ |
| | PDE-FIND | $f_t - 0.999f_{xx} + 1.264f - 1.863f^3 - 0.638ff_x - 0.081f^3f_x - 0.025f^2f_{xx} - 0.007ff_{xxx} - 0.001f^3f_{xxx} = 0$ |
| 6.25 | Adjoint | $f_t - f_{xx} = 0$ |
| | PDE-FIND | $f_t - 0.999f_{xx} + 2.769f - 4.051f^3 - 1.398ff_x - 0.185f^3f_x - 0.055f^2f_{xx} - 0.016ff_{xxx} - 0.002f^3f_{xxx} = 0$ |

## 4  PARTIAL OBSERVATIONS IN TIME

Here, we investigate how the error of the discovery task increases when only a subset of the fine data set is available. Consider the heat equation presented in section 3.1 and consider a data set created by solving the exact PDE using the Finite Difference method with $\Delta t = T/N_t$ where $N_t = 1000$ and $\Delta x = L/N_x$ and $N_x = 1000$.

Let us assume that we are only provided with a subset of this data set. As a test, let us take every $\nu$ time step as the input for the PDE discovery task, where $\nu \in \{1, 2, 4, 8, 16\}$. This corresponds to using $\{100, 50, 25, 12.5, 6.25\}\%$ of the total data set. By doing so, the accuracy of the Finite Difference method in estimating the time derivatives using the available data deteriorates, leading to large error in PDE discover task for PDE-FIND method.

However, the adjoint method can use a finer mesh in time compared to the data set in computing the forward and backward equations and only compare the solution to the data on the coarse mesh where data is available. We use $N_t = 1000$ for the forward and backward solvers in the adjoint method, and impose the final time condition where data is available. As shown in Table 1 and Fig. 2 the proposed adjoint method is able to recover the exact PDE regardless of how sparse the data set is in time.

We emphasize that while adjoint method can use a finer discretization in time than the one for data on $\mathcal{G}$ in solving forward and backward equations, it is bound to use similar or coarser spatial discretization as $\mathcal{G}$. This is due to the fact that the data points $\boldsymbol{f}^*$ are used for the initial condition of the forward model eq. equation 9, and the final condition of the backward adjoint equation eq. equation 8.

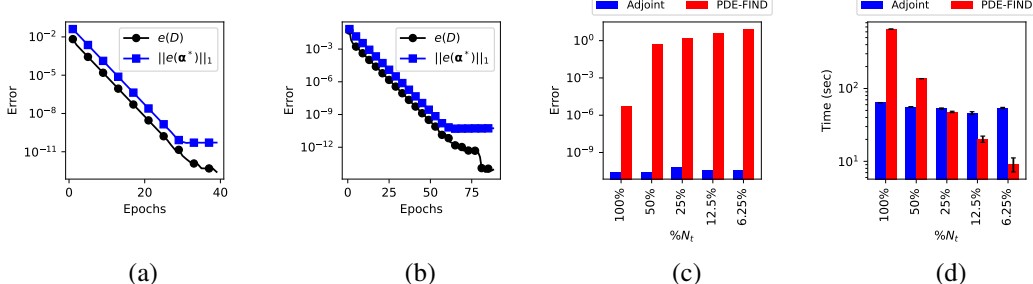

(a)    (b)    (c)    (d)

Figure 2: Evolution of the $L_1$-norm error in coefficients of all considered terms using adjoint method when only (a) 50% and (b) 6.25% (b) of the data set is available. Error and execution time of the adjoint method (blue) and PDE-FIND method (red) in finding the coefficients of true heat equation given sparse data set in time in (c) and (d).

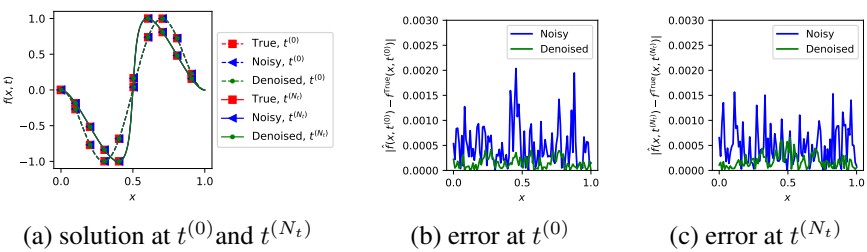

(a) solution at $t^{(0)}$ and $t^{(N_t)}$    (b) error at $t^{(0)}$    (c) error at $t^{(N_t)}$

Figure 3: Ground truth, noisy ($\sigma = 0.1\%$), and denoised solution to the Burgers' equation at the initial time $t^{(0)}$ and the final time $t^{(N_t)}$ where $N_t = 1000$ (a), as well as the error between noisy/denoised solution and the ground truth at these times in (b) and (c), respectively.

## 5 SENSITIVITY TO NOISE

Here, we investigate how the error increases once noise is added to the data set. In particular, we add noise $\epsilon \sim \mathcal{N}(0, \sigma^2)$ to each point of the data set for $\boldsymbol{f}^*$, where $\mathcal{N}(0, \sigma^2)$ denotes a normal distribution with zero mean and standard deviation $\sigma$. As test cases, we revisit the heat (section 3.1) and Burgers' equations (section B.1) with added noise of $\epsilon$ with $\sigma \in \{0.001, 0.005, 0.01, 0.1\}$ %. Before searching for the PDE, we first denoise the data set using Singular Value Decomposition and drop out terms with singular value below a threshold of $\mathcal{O}(10^{-4})$. As an example, we show the solution to Burgers' equation in the presence of $0.1\%$ noise in Figure 4.

As shown in Figure 4, adding noise to the data set deteriorates the accuracy in finding the correct coefficients of the underlying PDE for both the adjoint method and PDE-FIND method. We observe that the adjoint method, both with and without gradient averaging, is less susceptible to noise compared to PDE-FIND, albeit at a higher computational cost. Additionally, averaging the gradients in the adjoint method improves the accuracy around two orders of magnitude at higher computational cost.

## 6 ADDRESSING ILL-POSEDNESS

There may exist more than one PDE that replicates the data set. Therefore, the PDE discovery task is ill-posed due to the lack of uniqueness in the solution. This is an indication that further physically motivated constraints are needed to narrow the search space to find the desired PDE. However, among all possible PDEs, which PDE is found by the Adjoint method with the loss function defined as equation 3?

To answer this question, let us consider a simple example of the wave equation

$$f(x,t) = \sin(x - t) \tag{11}$$

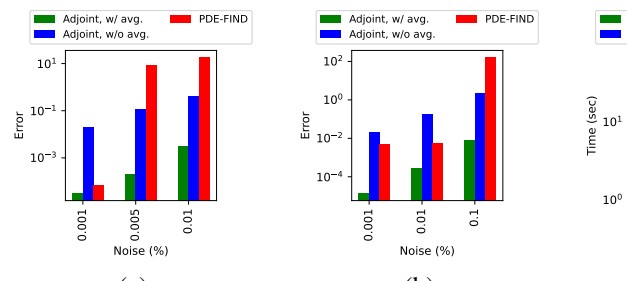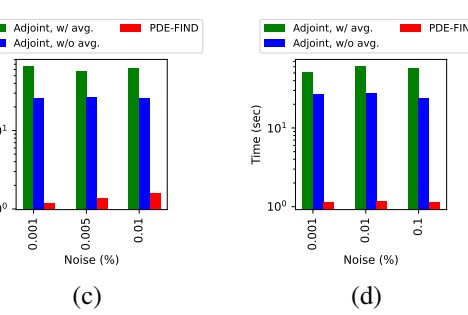

| (a) | (b) | (c) | (d) |

Figure 4: Error and execution time of the adjoint method with (green) and without averaging the gradients (blue), along with the PDE-FIND method (red) in finding the coefficients of the true PDE, i.e. heat equation (a)-(c) and Burgers' equation (b)-(d), given noisy data.

which is a solution to infinite PDEs. For example, one class of PDEs with solution $f$ is

$$f_t + kf_x + (k-1)f_{xxx} + c(f_{xx} + f_{xxxx}) = 0 \quad \forall k \in \mathbb{N} \text{ and } c \in \mathbb{R}, \tag{12}$$

defined in a domain $x \in [0, 2\pi]$ and $T = 1$. We create a data set using the exact $f$ on a grid with $N_t = 10$ time intervals and $N_x = 100$ spatial discretization points. Let us consider a similar setup as the heat equation example 3.1 with derivatives and polynomials indices $d \in \{1, ..., 6\}$ and $p = 1$ as the initial guess for the forward model. This leads to 6 terms with unknown coefficients $\boldsymbol{\alpha}$. Here, we enable averaging and use a finer discretization in time (100 steps for forward and backward solvers in each time interval) to cope with the instabilities of the Finite Difference solver due to the inclusion of the high-order derivatives. We also disable thresholding except at the end of the algorithm.

The proposed Adjoint method returns the solution

$$f_t + 0.996f_x = 0 \tag{13}$$

which is the PDE with the least number of terms compared to all possible PDEs. We note that for the same problem setting, PDE-FIND finds

$$f_t + 0.9897f_x = 0. \tag{14}$$

The identified form of PDE can be explained by the use of regularization term in the cost function 3, which enforces the minimization of the PDE coefficients. Clearly, the regularization term may be changed to find other possible solutions of this ill-posed problem.

## 7 DISCUSSION

Below we highlight and discuss strengths and weaknesses of the proposed adjoint method.

**Strengths.** The proposed method has several strengths:

1. The proposed adjoint-based method of discovering PDEs can provides coefficients of the true governing equation with significant accuracy.

2. Since the gradient of the cost function with respect to parameters are derived analytically, the optimization problem converges fast. In particular, the adjoint method becomes cheaper than PDE-FIND as the size of the data set increases. The adjoint method by construction finds the optimal relation between gradient of the cost function and the error in the data points. This was achieved by finding the extremum of the objective functional using variational derivative. We note that a clear difference from the point-wise loss $||f - f^*||_2$ equipped with backprobagation used in PDE-FIND method is that the adjoint method weights the error at discrete points with the Lagrange multipliers, see Eq. equation 4 and the final condition Eq. equation 8.

3. Since the adjoint method uses a PDE solver to find the underlying governing equation, there is a guarantee that the found PDE can be solved numerically with the same PDE solver as the one used by the adjoint method.

4. The adjoint method can use a finer mesh in time compared to the available discretization of the data set. This allows an accurate recovery of the underlying PDE compared to the PDE-FIND, where the error in the latter increases as the data set gets coarser since it estimates derivatives directly (either with Finite Difference or a polynomial fit) using the given data set.

**Weaknesses.**    Our proposed method has some limitations:

1. In order to use the proposed adjoint method for discovering PDEs, a general solver of PDEs needs to be implemented. Here, we used Finite Differences which can be replaced with more advance solvers. Clearly, the proposed adjoint method is most effective when there is a prior knowledge of the underlying PDE form, and an appropriate numerical solver is deployed.We note that an inherent limitation of the proposed adjoint method is the possibility of encountering either ill-posed forward or backward equations during optimization, which limits the time step size.

2. In this work, we used the same spatial discretization as the input data. If the spatial grid of input data is too coarse for the PDE solver, one has to use interpolation to estimate the data on a finer spatial grid that is more appropriate for the PDE solver.

3. In this work, we made the assumption that the underlying PDE can be solved numerically. This can be a limitation when there are no stable numerical methods to solve the true PDE. In this scenario, the proposed method may find another PDE that is solvable and fits to the data with a notable error.

## 8    CONCLUSION

In this work, we introduce a novel mathematical method for the discovery of partial differential equations given data using the adjoint method. By formulating the optimization problem in the variational form using the method of Lagrange multipliers, we find an analytic expression for the gradient of the cost function with respect to the parameters of the PDE as a function of the Lagrange multipliers and the forward model estimate. Then, using variational calculus, we find a backwards-in-time evolution equation for the Lagrange multipliers which incorporates the error with a source term (the adjoint equation). Hence, we can use the same solver for both forward model and the backward Lagrange equations. Here we used Finite Differences to estimate the spatial derivatives and forward Euler for the time derivatives, which indeed can be replaced with more stable and advanced solvers.

We compared the proposed adjoint method against PDE-FIND on several test cases. While PDE-FIND seems to be faster for small size problems, we observe that the adjoint method equipped with forward/backward solvers becomes faster than PDE-FIND as the size of the data set increases. Also, the adjoint method can provide machine-accuracy in identifying and finding the coefficients when the data set is noise-free. Furthermore, in the case of discovering PDEs for PDFs given its samples, both methods seem to suffer enormously form noise/bias associated with the finite number of samples and the Finite Difference on histogram. This motivates the use of smooth and least biased density estimator in these methods such as Tohme et al. (2023) in future work. In the future work, we intend to combine the adjoint-based method for the discovery of PDE with PINNs as the solver instead of Finite Difference method. This would allow us to handle noisy and sparse data as well as deploying larger time steps in estimating the forward and backward solvers.

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

## A   FLOWCHARTS OF ADJOINT METHOD

Here, we present flowcharts to illustrate the proposed adjoint algorithms 1-2 with and without averaging the gradients in Fig. 5.

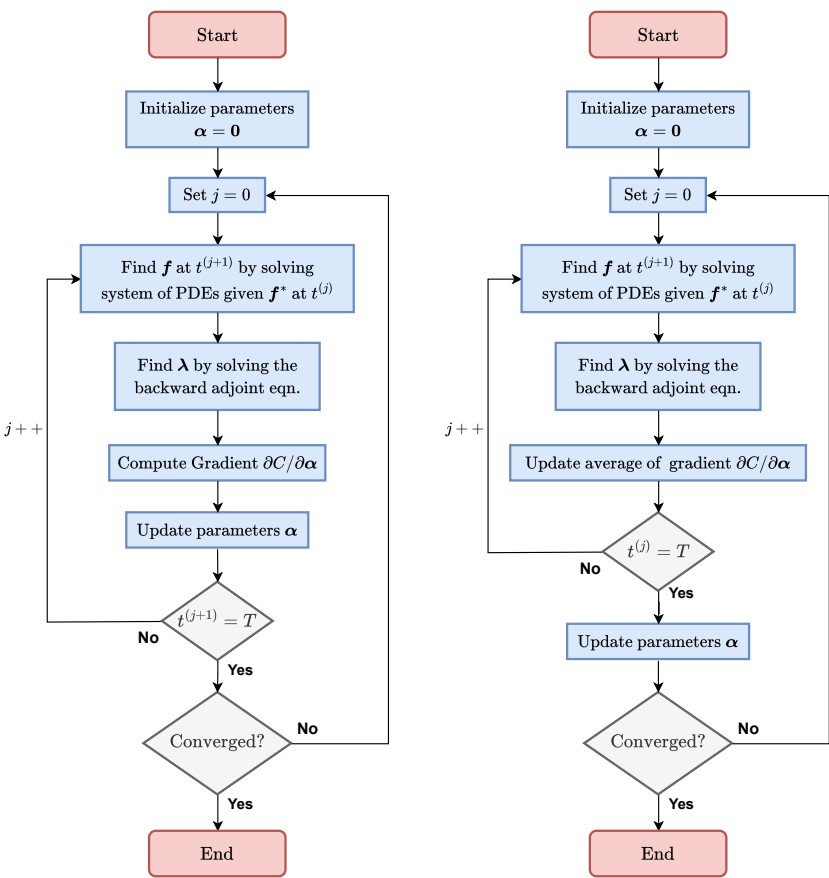

Figure 5: Training flowchart of the Adjoint method in finding PDEs (left) without and (right) with gradient averaging.

## B   GALLERY OF EXAMPLES

Besides the heat equation 3.1, here we show how the proposed adjoint method for discovering governing equation performs in finding Burgers', Kuramoto Sivashinsky, Random Walk, Reaction Diffusion System of Equations, and wave equation.

### B.1   BURGERS' EQUATION

As a nonlinear test case, let us consider the data from Burgers' equation given by

$$\frac{\partial f}{\partial t} + \frac{\partial (Af^2)}{\partial x} = 0 \tag{15}$$

where $A = -1$. The data is obtained with similar simulation setup as for heat equation (Section 3.1) except for the time step, i.e. $\Delta t = 0.05\Delta x/(1 + |A|)$.

Similar to Section 3.1, we adopt a system of one PDE with one-dimensional input. We also consider derivatives and polynomials with indices $d, p \in \{1, 2, 3\}$ in the construction of the forward model. This leads to 9 terms whose coefficients we find using the proposed adjoint method. As shown in

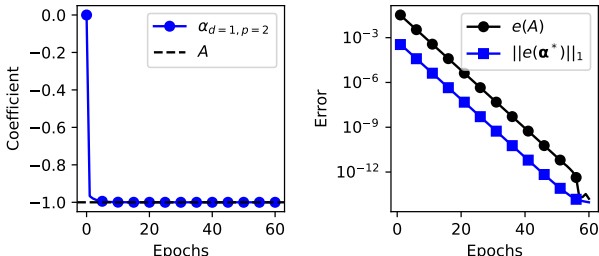

Figure 6: The estimated coefficient corresponding to $A$ (left) and the $L_1$-norm error of all considered coefficients (right) given the discretized data of Burgers' equation during training.

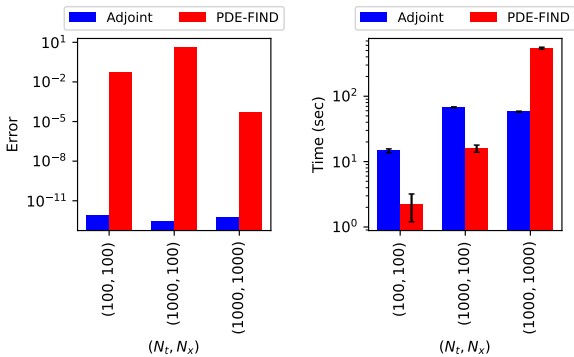

Figure 7: $L_1$-norm error of the estimated coefficients (left) and the execution time (right) for discovering the Burgers' equation equation using the Adjoint method (blue) and PDE-FIND method (red), given data on a grid with $N_t \in \{100, 1000\}$ steps in $t$, and $N_x \in \{100, 1000\}$ nodes in $x$.

Fig. 6, the proposed adjoint method finds the correct coefficients, i.e. $\alpha_{d=1,p=2}$ that corresponds to $D$ as well as all the irrelevant ones denoted by $\boldsymbol{\alpha}^*$, up to machine accuracy in $\mathcal{O}(10)$ epochs.

Next, we compare the solution obtained from the adjoint method to the one from PDE-FIND using STRidge optimization method. Here, we compare the error on coefficients and computational time between the adjoint and PDE-FIND by repeating the task for the data set with increasing size, i.e. $(N_t, N_x) \in \{(100, 100),\ (1000, 100),\ (1000, 1000)\}$. As depicted in Fig. 7, the adjoint method provides us with more accurate solution across the different discretization sizes. Regarding the computational cost, while PDE-FIND seems faster on smaller data sets, as the size of the data grows, it becomes increasingly more expensive than the adjoint method. Similar to the heat equation, for the mesh size $(N_t, N_x) = (1000,\ 1000)$ we obtain one order of magnitude speed up compared to PDE-FIND.

### B.2 KURAMOTO SIVASHINSKY EQUATION

As a more challenging test case, let us consider the recovery of the Kuramoto-Sivashinsky (KS) equation given by

$$\frac{\partial f}{\partial t} + A\frac{\partial f^2}{\partial x} + B\frac{\partial^2 f}{\partial x^2} + C\frac{\partial^4 f}{\partial x^4} = 0 \tag{16}$$

where $A = -1$, $B = 0.5$ and $C = -0.5$. The data is generated similar to previous sections except for the grid $(N_t, N_x) = (64, 256)$ and the time step size $\Delta t = 0.01\Delta x^4/(1 + |C|)$.

Here again, we adopt a system of one PDE with one-dimensional input. As a guess for the forward model, we consider terms consisting of derivatives with indices $d \in \{1, 2, 3, 4\}$ and polynomials with indices $b \in \{1, 2\}$, leading to 8 terms whose coefficients we find using the proposed adjoint method. As shown in Fig. 8, the adjoint method finds the coefficient with error of $\mathcal{O}(10^{-5})$, yet achieving machine accuracy seems not possible.

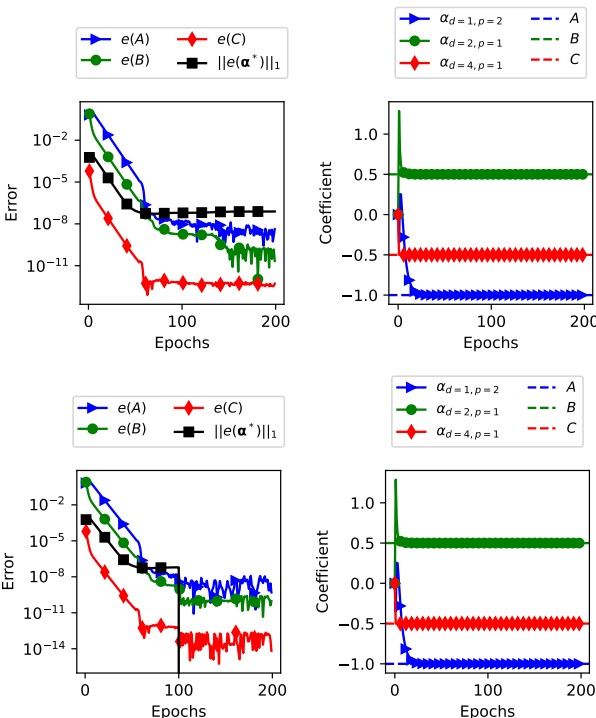

Figure 8: The estimated coefficients corresponding to $A, B, C$ (left) and the $L_1$-norm error of all considered coefficients (right) given the discretized data of the KS equation during training without (top) and with (bottom) active thresholding for $\mathrm{Epochs} > N_{\mathrm{thr}} = 100$.

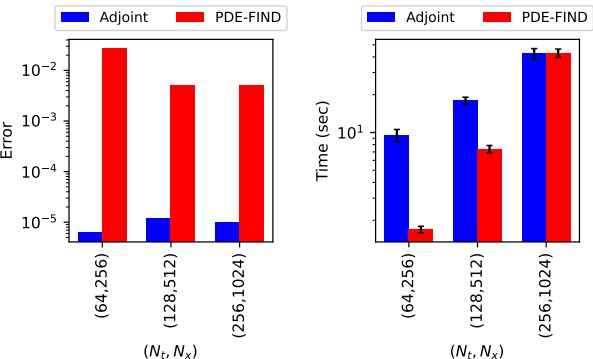

Figure 9: $L_1$-norm error of the estimated coefficients (left) and the execution time (right) for discovering the KS equation using the Adjoint method (blue) and PDE-FIND method (red), given data on a grid with $N_t \in \{64, 128, 256\}$ steps in $t$, $N_x \in \{256, 512, 1024\}$ nodes in $x$.

Again, in Fig. 9 we make a comparison between the predicted PDE using the adjoint method against PDE-FIND. In particular, we consider a data set on a temporal/spatial mesh of size $(N_t, N_x) = \{(64, 256), (128, 512), (256, 1024)\}$ and compare how the error and computational cost vary. Similar to previous sections, the error is reported by comparing the obtained coefficients against the coefficients of the exact PDE in $L_1$-norm. Interestingly, the PDE-FIND method has 3 to 4 orders of magnitude larger error compared to the adjoint method. Also, in terms of cost, the training time for PDE-FIND seems to grow at a higher rate than the adjoint method as the (data) mesh size increases.

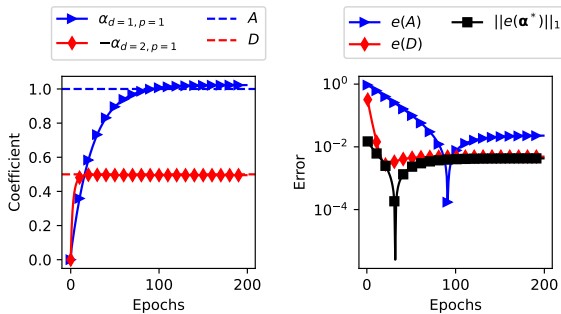

Figure 10: The estimated coefficients corresponding to $A$ and $D$ (left) and the $L_1$-norm error of all considered coefficients (right) of the Fokker-Planck equation as the governing law for the PDF associated with the random walk during training.

## B.3 Random Walk

Next, let us consider the recovery of the governing equation on probability density function (PDF) given samples of its underlying stochastic process. As an example, we consider the Itô process

$$dX = A dt + \sqrt{2D} dW \tag{17}$$

where $A = 1$ is drift and $D = 0.5$ is the diffusion coefficient, and $W$ denotes the standard Wiener process with $\mathrm{Var}(dW) = \Delta t$. We generate the data set by simulating the random walk using Euler-Maruyama scheme starting from $X(t = 0) = 0$ for $N_t = 50$ steps with a time step size of $\Delta t = 0.01$. We estimate the PDF using histogram with $N_x = 100$ bins and $N_s = 1000$ samples.

Let us denote the distribution of $X$ by $f$. Itô's lemma gives us the Fokker-Planck equation

$$\frac{\partial f}{\partial t} + A \frac{\partial f}{\partial x} - D \frac{\partial^2 f}{\partial x^2} = 0 \,. \tag{18}$$

Given the data set for $f$ on a mesh of size $(N_t, N_x)$, we can use Finite Difference to compute the contributions from derivatives of $f$ in the governing law. Since this is one of the challenging test cases due to noise, here we only consider three possible terms in the forward model, consisting of derivatives with indices $d \in \{1, 2, 3\}$ and polynomial power $p = 1$. In Fig. 10, we show how the error of finding the correct coefficients evolves during training for the adjoint method. Clearly, the adjoint method seems to recover the true PDE with $L_1$ error of $\mathcal{O}(10^{-2})$ in its coefficients.

In Fig. 11, we make a comparison with PDE-FIND for the same number and order of terms as the initial guess for the PDE. We compare the two methods for a range of grid and sample sizes, i.e. $N_t \in \{50, 100\}$, $N_x = 100$, and $N_s \in \{10^3, 10^4\}$. It turns out that the proposed adjoint method overall provides more accurate estimate of the coefficients than PDE-FIND, though at a higher cost. In Table 2, we show the discovered PDEs for both methods across the different discretizations.

Table 2: Recovery of the Fokker-Planck equation, i.e. $f_t + f_x - 0.5 f_{xx} = 0$, using the proposed adjoint method against PDE-FIND method given samples of the underlying stochastic process for various discretization parameters.

| $N_t$ | $N_x$ | $N_s$ | Method | Recovered PDE |
|---|---|---|---|---|
| 50 | 100 | 1000 | Adjoint | $f_t + 1.025 f_x - 0.465 f_{xx} = 0$ |
| | | | PDE-FIND | $f_t + 0.798 f_x - 0.454 f_{xx} = 0$ |
| | | 10000 | Adjoint | $f_t + 1.022 f_x - 0.495 f_{xx} = 0$ |
| | | | PDE-FIND | $f_t + 0.818 f_x - 0.496 f_{xx} = 0$ |
| 100 | 100 | 1000 | Adjoint | $f_t + 1.010 f_x - 0.543 f_{xx} = 0$ |
| | | | PDE-FIND | $f_t + 0.863 f_x - 0.560 f_{xx} = 0$ |
| | | 10000 | Adjoint | $f_t + 1.015 f_x - 0.589 f_{xx} = 0$ |
| | | | PDE-FIND | $f_t + 0.894 f_x - 0.612 f_{xx} = 0$ |

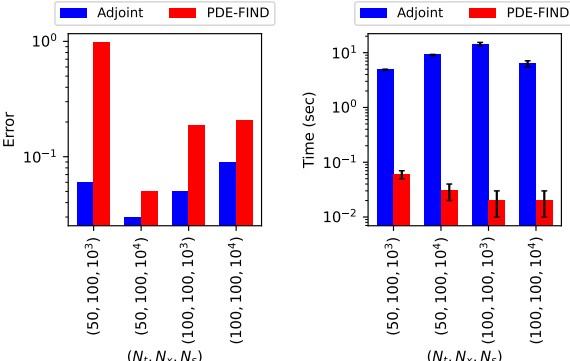

Figure 11: $L_1$-norm error of the estimated coefficients (left) and the execution time (right) for discovering the Fokker-Planck equation using the proposed Adjoint method (blue) and PDE-FIND method (red), given samples of its underlying stochastic process with $N_t \in \{50, 100\}$ steps in $t$, $N_x = 100$ histogram bins, and $N_s \in \{10^3, 10^4\}$ samples.

### B.4 REACTION DIFFUSION SYSTEM OF EQUATIONS

In order to show scalability and accuracy of the adjoint method for a system of PDEs in a higher dimensional space, let us consider a system of PDEs given by

$$\frac{\partial u}{\partial t} + c_0^u \nabla_{x_1}^2 [u] + c_1^u \nabla_{x_2}^2 [u] + R^u(u, v) = 0, \tag{19}$$

$$\frac{\partial v}{\partial t} + c_0^v \nabla_{x_1}^2 [v] + c_1^v \nabla_{x_2}^2 [v] + R^v(u, v) = 0 \tag{20}$$

where

$$R^u(u, v) = c_2^u u + c_3^u u^3 + c_4^u u v^2 + c_5^u u^2 v + c_6^u v^3 \tag{21}$$

$$R^v(u, v) = c_2^v v + c_3^v v^3 + c_4^v v u^2 + c_5^v v^2 u + c_6^v u^3 \tag{22}$$

We construct the data set by solving the system of PDEs Eqs. equation 19-equation 20 using a 2nd order Finite Difference scheme with initial values

$$u_0 = a \sin\left(\frac{4\pi x_1}{L_1}\right) \cos\left(\frac{3\pi x_2}{L_2}\right) \left(L_1 x_1 - x_1^2\right) \left(L_2 x_2 - x_2^2\right)$$

$$v_0 = a \cos\left(\frac{4\pi x_1}{L_1}\right) \sin\left(\frac{3\pi x_2}{L_2}\right) \left(L_1 x_1 - x_1^2\right) \left(L_2 x_2 - x_2^2\right)$$

where $a = 100$, and the coefficients

$$\boldsymbol{c}^u = [c_i^u]_{i=0}^6 = [-0.1, -0.2, -0.3, -0.4, 0.1, 0.2, 0.3]$$

$$\boldsymbol{c}^v = [c_i^v]_{i=0}^6 = [-0.4, -0.3, -0.2, -0.1, 0.3, 0.2, 0.1].$$

We generate data by solving the system of PDEs Eqs. (19)-(20) using the Finite Difference method and forward Euler scheme for $N_t = 25$ steps with a time step size of $\Delta t = 10^{-6}$, and in the domain $\Omega = [0, L_1] \times [0, L_2]$ where $L_1 = L_2 = 1$ which is discretized using a uniform grid with $N_{x_1} \times N_{x_2} = 50^2$ nodes leading to mesh size $\Delta x_1 = \Delta x_2 = 0.02$. In Fig. 12 we show the solution to the system at time $T = N_t \Delta t$ for $u$ and $v$.

We consider a system consisting of two PDEs, i.e. $N = \dim(\boldsymbol{f}) = \dim(\boldsymbol{p}) = 2$, with two-dimensional input, i.e. $n = \dim(\boldsymbol{x}) = \dim(\boldsymbol{d}) = 2$. Here, $\dim(\boldsymbol{f}) = \dim(\mathrm{Ima}(\boldsymbol{f}))$, where $\mathrm{Ima}(\cdot)$ denotes the image (or output) of a function.

In order to use the developed adjoint method, we construct a guess forward system of PDEs (or forward model) using derivatives up to 2nd order and polynomials of up to 3rd order. That is, $d_{\max} = 2$ and $p_{\max} = 3$. This leads to 90 terms whose coefficients we find using the proposed

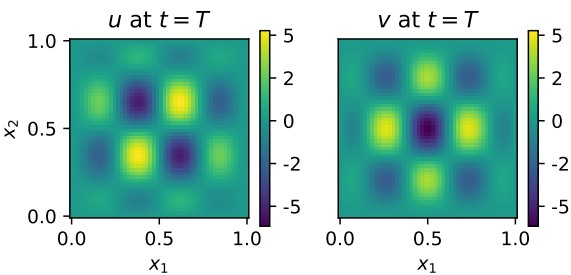

Figure 12: Solution to the reaction diffusion system of PDEs at time $t = T$ for $u$ (left) and $v$ (right).

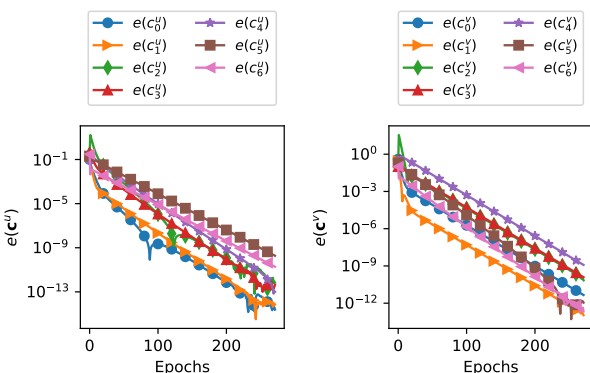

Figure 13: $L_1$-norm error in the estimated coefficients of the reaction diffusion system of PDEs during training.

adjoint method (an illustrative derivation of the candidate terms can be found in Appendix F.2). The solution to the constructed model $\boldsymbol{f} \approx [u, v]$ as well as the adjoint equation for $\boldsymbol{\lambda}$ is found using the same discretization as the data set.

As shown in Fig. 13, the adjoint method finds the correct equations with error up to $\mathcal{O}(10^{-12})$. Furthermore, the coefficients corresponding to the irrelevant terms $\boldsymbol{\alpha}^*$ tend to zero with error of $\mathcal{O}(10^{-11})$, see Fig. 14.

Furthermore, we have compared the adjoint method against PDE-FIND for a range of grid sizes in Fig. 16. We observe that the cost of PDE-FIND grows with higher rate than adjoint method as the size of the data set increases.

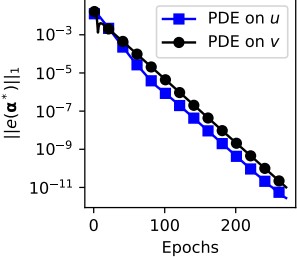

Figure 14: $L_1$-norm error in the estimated coefficients of the irrelevant terms compared to the true reaction diffusion system of PDEs during training, i.e. $||e(\boldsymbol{\alpha}^*)||_1 = ||\boldsymbol{\alpha}^*||_1$.

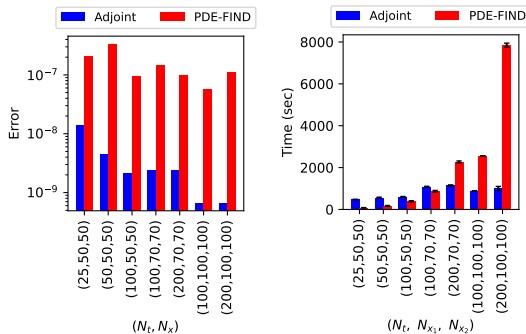

Figure 16: Comparing the error and execution time of the adjoint method (blue) to PDE-FIND method (red) against the size of the data set for the tolerance of $10^{-7}$ in the discovered coefficients.

## B.5 WAVE EQUATION

Consider wave equation

$$f(x, t) = \sin(x - t) \tag{23}$$

which is a solution to infinite PDEs. For example, one class of PDEs with solution $f$ is

$$f_t + k f_x + (k - 1) f_{xxx} + c(f_{xx} + f_{xxxx}) = 0 \quad \forall k \in \mathbb{N} \text{ and } c \in \mathbb{R}, \tag{24}$$

defined in a domain $x \in [0, 2\pi]$ and $T = 1$. We create a data set using the exact $f$ on a grid with $N_t = 10$ time intervals and $N_x = 100$ spatial discretization points.

Let us consider a similar setup as the heat equation example 3.1 with derivatives and polynomials indices $d \in \{1, ..., 6\}$ and $p = 1$ as the initial guess for the forward model. This leads to 6 terms with unknown coefficients $\boldsymbol{\alpha}$. Here, we enable averaging and use a finer discretization in time (100 steps for forward and backward solvers in each time interval) to cope with the instabilities of the Finite Difference solver due to the inclusion of the high-order derivatives. We also disable thresholding except at the end of the algorithm.

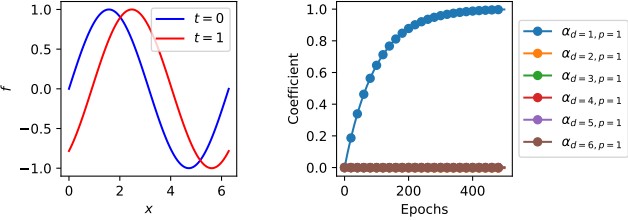

Figure 15: Profile of $f$ at $t = 0$ and $t = 1$ (left) and the evolution of considered coefficients during adjoint optimization (right)

As shown in Fig. 15, the proposed Adjoint method returns the solution

$$f_t + 0.996 f_x = 0 \tag{25}$$

which is the PDE with the least number of terms compared to all possible PDEs. We note that for the same problem setting, PDE-FIND identifies the same form of the PDE, i.e.

$$f_t + 0.9897 f_x = 0 . \tag{26}$$

## C INCOMPLETE GUESSED PDE SPACE

In this section, we investigate the outcome of the adjoint method when the exact terms are not included in the initial guessed PDE form. Here, we define the space of PDE where the exact terms

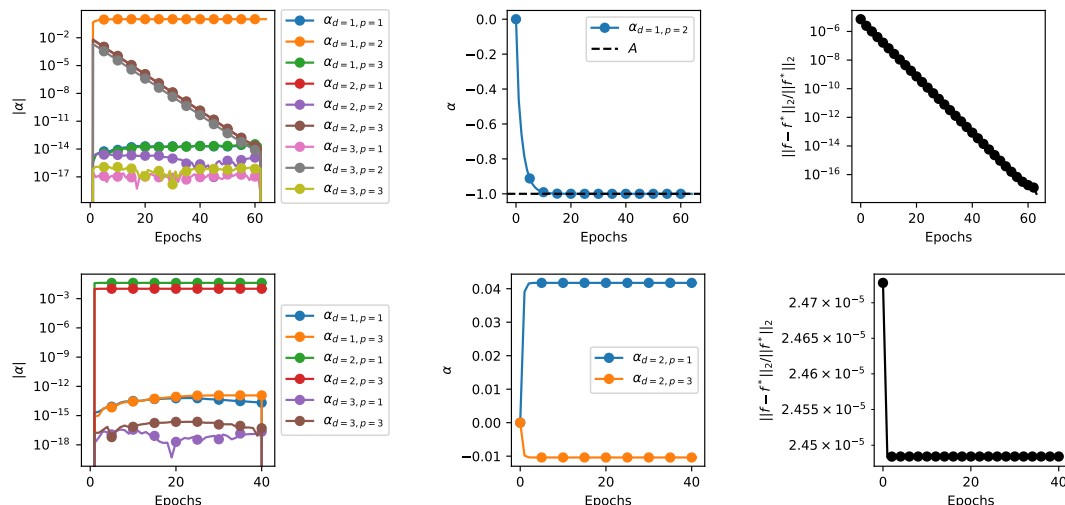

Figure 17: The adjoint method applied to the Burgers' equation for complete (top) and incomplete (bottom) space of guessed PDEs.

are included in the general forward model equation 1 as complete. If the considered general form of PDE equation 1 does not include all the terms of the exact PDE, we denote that as an incomplete guessed PDE space.

Let us take the data from the numerical solution to Burgers' equation used in section B.1 with discretization $N_t = N_x = 100$. For the complete forward model, we again consider derivatives and polynomials with indices $d, p \in \{1, 2, 3\}$ in the construction of the forward model. This leads to 9 terms whose coefficients we find using the proposed adjoint method. For the incomplete space of PDE, we take derivatives and polynomials with indices as $d \in \{1, 2, 3\}$ and $p \in \{1, 3\}$, leading to 6 terms. Clearly, the incomplete guessed PDE space does not include the term $\alpha_{d=1,p=2}\partial f^2/\partial x$. Now, we would like to see which PDE is returned by the adjoint method.

In Figure 17, we made a comparison between the evolution of coefficients and $L_2$ norm error of the estimated forward model against the data. While the complete space monotonically converges to the exact solution up to machine accuracy, the incomplete space of PDE delivers another PDE, i.e.

$$\frac{\partial f}{\partial t} + \frac{\partial^2}{\partial x^2}(0.04f - 0.01f^3) = 0, \tag{27}$$

with the relative $L_2$ error of $\mathcal{O}(10^{-5})$ between forward model estimation and the data points. The fact that the $L_2$ error between $f$ and $f^*$ does not decrease is an indication that the considered space of PDE is incomplete and additional terms must be included. We note that here we assumed there is no noise in the data set. However, in the presence of noise, the $L_2$ error between $f$ and $f^*$ may stagnate at the noise level, which makes the analysis on the completeness of the PDE space more challenging.

## D   IMPACT OF HYPERPARAMETERS ON ADJOINT METHOD

In this section, we study the impact of some of the hyperparameters used in the adjoint algorithm. We repeat the PDE discovery experiment for Burgers's and Kuramoto Sivashinsky equation with data on a grid with $N_x = N_t = 100$, as described in B.1 and B.2. We check the error in the outcome coefficients and the solution of estimated forward model compared to the data.

As shown in Figure 18, by increasing the regularization factor $\epsilon_0$, the optimization problem seems to converge faster to a stationary solution. In case of Burgers' equation, we considered $\epsilon_0 \in \{10^{-4}, 10^{-8}, 10^{-12}, 10^{-16}\}$, where for all values of $\epsilon_0$ the exact solution is recovered. However, in the case of Kuramoto Sivashinsky with $\epsilon_0 \in \{10^{-10}, 10^{-12}, 10^{-14}, 10^{-16}\}$, the solution seems to be more sensitive to $\epsilon_0$. Here, we fix the other hyperparameters $\gamma_{\text{thr}} = 10^{-16}$ and $\beta = 2 \times 10^{-3}$

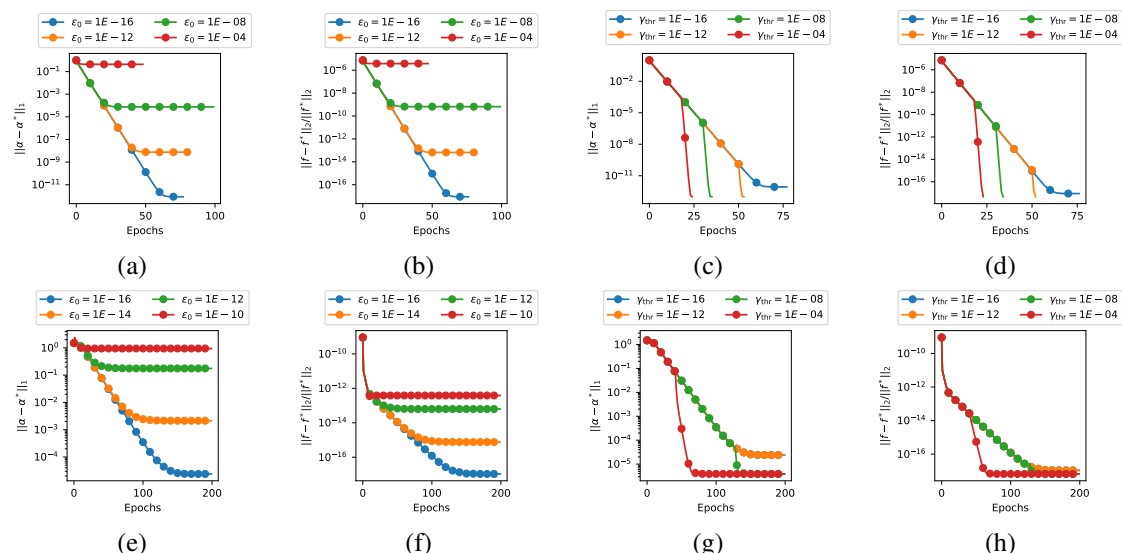

(a)      (b)      (c)      (d)

(e)      (f)      (g)      (h)

Figure 18: Impact of regularization factor $\epsilon_0$ and thresholding tolerance $\gamma_{\text{thr}}$ on the error of adjoint method for Burgers' equation (a-d) and Kuramoto Sivashinsky equation (e-h).

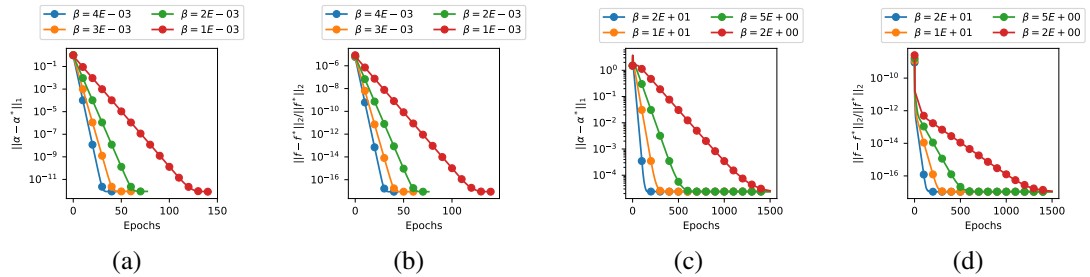

(a)      (b)      (c)      (d)

Figure 19: Impact of the free parameter $\beta$ in the learning rate on the error of adjoint method for Burgers' equation (a-b) and Kuramoto Sivashinsky equation (c-d).

for the Burgers's equation and $\beta = 20$ for Kuramoto Sivashinsky equation. We observe that high regularization factor deteriorates the accuracy, while stabilizing the regression problem.

Next, we investigate how the error changes with the thresholding tolerance where $\gamma_{\text{thr}} \in \{10^{-4}, 10^{-8}, 10^{-12}, 10^{-16}\}$. Here, we fix the other hyperparameters $\epsilon_0 = 10^{-16}$ and $\beta = 2 \times 10^{-3}$ for the Burgers's equation and $\beta = 20$ for Kuramoto Sivashinsky equation. Although using smaller $\gamma_{\text{thr}}$ allows faster convergence to a stationary solution almost in all cases, we remind the reader that $\gamma_{\text{thr}}$ should be large enough to allow enough training of the coefficients before truncating terms. In other words, the user should avoid trivial scenarios where the initial guess for coefficients $\boldsymbol{\alpha}$ are zero and the thresholding is applied from very beginning of the training.

Finally, we show the impact of the free parameter $\beta$ in the learning rate on the resulting PDE discovered by the adjoint method. We compared the solution of adjoint method using $\beta \in \{10^{-3}, 2 \times 10^{-3}, 3 \times 10^{-3}, 4 \times 10^{-3}\}$ for the Burgers' equation, and $\beta \in \{2, 5, 10, 20\}$ for the Kuramoto Sivashinsky equation. Also, we fix the other hyperparameters $\gamma_{\text{thr}} = \epsilon_0 = 10^{-16}$. As shown in Figure 19, regardless of the value of $\beta$, adjoint method delivers the same solution. However, larger values of $\beta$ lead to faster convergence to the solution, if the numerical solver does not become unstable. The upper bound of $\beta$ is limited by the stability of the guessed PDE, and can be found with try-and-error.

# E  Justification for the Choice of the Learning Rate

In the proposed adjoint method, we considered the update rule

$$\alpha_{i,\boldsymbol{d},\boldsymbol{p}} \leftarrow \alpha_{i,\boldsymbol{d},\boldsymbol{p}} - \eta \frac{\partial \mathcal{C}}{\partial \alpha_{i,\boldsymbol{d},\boldsymbol{p}}} \tag{28}$$

for $i = 1, \ldots, N$. Here, we given a justification for our choice of the learning parameter $\eta$.

From the expression for the gradient of cost function with respect to parameters equation 4, i.e.

$$\frac{\partial \mathcal{C}}{\partial \alpha_{i,\boldsymbol{d},\boldsymbol{p}}} = (-1)^{|\boldsymbol{d}|} \frac{1}{\Delta \boldsymbol{x} \Delta t} \int \boldsymbol{f}^{\boldsymbol{p}} \nabla_{\boldsymbol{x}}^{(\boldsymbol{d})} [\lambda_i] d\boldsymbol{x} dt + 2\epsilon_0 \alpha_{i,\boldsymbol{d},\boldsymbol{p}}, \tag{29}$$

we can see that

$$\left| \frac{\partial \mathcal{C}}{\partial \alpha_{i,\boldsymbol{d},\boldsymbol{p}}} \right| = \mathcal{O}(\nabla_{\boldsymbol{x}}^{(\boldsymbol{d})}) \tag{30}$$

$$\leq \mathcal{O}(h^{-|\boldsymbol{d}|}) \tag{31}$$

where $h = \min(\Delta \boldsymbol{x})$. So, the magnitude of the gradient scales exponentially with the order of the derivative $d$. The highest order terms, i.e. the terms with $d = d_{\max} = \max(|\boldsymbol{d}|)$, have the largest magnitude for their gradients. This means that by taking a constant learning rate $\eta$, the adjoint method would find the coefficients of the highest order terms first. This effect leads to the non-uniform convergence of the adjoint method.

In order to enforce uniform convergence on all PDE parameters, in this paper we consider

$$\eta = \beta \min(\Delta \boldsymbol{x})^{|\boldsymbol{d}| - d_{\max}} \tag{32}$$

as the learning rate which encodes the scaling with respect to the order of derivative for each PDE term. With this choice of learning rate, we have

$$\eta \left| \frac{\partial \mathcal{C}}{\partial \alpha_{i,\boldsymbol{d},\boldsymbol{p}}} \right| \leq \mathcal{O}(h^{|\boldsymbol{d}| - d_{\max}}) \mathcal{O}(h^{-|\boldsymbol{d}|}) \tag{33}$$

$$\leq \mathcal{O}(h^{-d_{\max}}), \tag{34}$$

for all $i, \boldsymbol{d}, \boldsymbol{p}$. Hence, our choice of $\eta$, i.e. Eq. equation 32, enforces uniform convergence on all PDE parameters.

# F  Illustrative Derivation of the Adjoint Equations for the Considered Cases.

Although the proposed method and its algorithm can be and has been computed in an automated fashion, here we show two detailed illustrative examples for 1-dimensional and 2-dimensional cases presented in Section 3 for the sake of better understanding the used notation and how the library of candidate terms looks like.

## F.1  Heat and Burgers' Equations

As mentioned in Sections 3.1 and B.1, for these two cases, we consider a system consisting of a single PDE, i.e. $N = \dim(\boldsymbol{f}) = \dim(\boldsymbol{p}) = 1$ where $\boldsymbol{f} = f$ and $\boldsymbol{p} = p$, in a one-dimensional input space, i.e. $n = \dim(\boldsymbol{x}) = \dim(\boldsymbol{d}) = 1$ where $\boldsymbol{x} = x$, and $\boldsymbol{d} = d$. In addition, we consider candidate terms consisting of derivatives with indices $d \in \{1, 2, 3\}$ and polynomials with indices $p \in \{1, 2, 3\}$).

In other words, $d_{\max} = 3$ and $p_{\max} = 3$. The resulting forward model in Eq. 1 takes the form

$$
\begin{aligned}
\mathcal{L}[f] &= \frac{\partial f}{\partial t} + \sum_{d=1}^{3}\sum_{p=1}^{3}\alpha_{d,p}\frac{\partial^d\big(f^p\big)}{\partial x^d} \\
&= \frac{\partial f}{\partial t} + \alpha_{1,1}\frac{\partial f}{\partial x} + \alpha_{1,2}\frac{\partial\big(f^2\big)}{\partial x} + \alpha_{1,3}\frac{\partial\big(f^3\big)}{\partial x} \\
&\quad + \alpha_{2,1}\frac{\partial^2 f}{\partial x^2} + \alpha_{2,2}\frac{\partial^2\big(f^2\big)}{\partial x^2} + \alpha_{2,3}\frac{\partial^2\big(f^3\big)}{\partial x^2} \\
&\quad + \alpha_{3,1}\frac{\partial^3 f}{\partial x^3} + \alpha_{3,2}\frac{\partial^3\big(f^2\big)}{\partial x^3} + \alpha_{3,3}\frac{\partial^3\big(f^3\big)}{\partial x^3}
\end{aligned}
\tag{35}
$$

where $\alpha_{d,p}$ denotes the parameter corresponding to the term with $d$-th derivative and $p$-th polynomial order. As we can observe, we have 9 terms with unknown coefficients $\boldsymbol{\alpha} = [\alpha_{d,p}]_{d\in\{1,2,3\},p\in\{1,2,3\}}$ that we aim to find using the proposed adjoint method.

The cost functional in this case is simply

$$
\mathcal{C} = \sum_{j,k}\Big(f^*\big(x^{(k)},t^{(j)}\big) - f\big(x^{(k)},t^{(j)}\big)\Big)^2 + \frac{1}{\Delta x\Delta t}\int\lambda(x,t)\mathcal{L}[f(x,t)]dxdt + \epsilon_0||\boldsymbol{\alpha}||_2^2 . \tag{36}
$$

Letting variational derivatives of $\mathcal{C}$ with respect to $f$ to be zero, and using integration by parts, the corresponding adjoint equation can be obtained as

$$
\begin{aligned}
\frac{\partial\lambda}{\partial t} &= \sum_{d=1}^{3}\sum_{p=1}^{3}(-1)^d\alpha_{d,p}\frac{\partial\big(f^p\big)}{\partial f}\frac{\partial^d\lambda}{\partial x^d} \\
&= -\alpha_{1,1}\frac{\partial\lambda}{\partial x} - \alpha_{1,2}\big(2f\big)\frac{\partial\lambda}{\partial x} - \alpha_{1,3}\big(3f^2\big)\frac{\partial\lambda}{\partial x} \\
&\quad + \alpha_{2,1}\frac{\partial^2\lambda}{\partial x^2} + \alpha_{2,2}\big(2f\big)\frac{\partial^2\lambda}{\partial x^2} + \alpha_{2,3}\big(3f^2\big)\frac{\partial^2\lambda}{\partial x^2} \\
&\quad - \alpha_{3,1}\frac{\partial^3\lambda}{\partial x^3} - \alpha_{3,2}\big(2f\big)\frac{\partial^3\lambda}{\partial x^3} - \alpha_{3,3}\big(3f^2\big)\frac{\partial^3\lambda}{\partial x^3}
\end{aligned}
\tag{37}
$$

with final condition $\lambda(x^{(k)},t^{(j+1)}) = 2(f^*(x^{(k)},t^{(j+1)}) - f(x^{(k)},t^{(j+1)}))$ for all $j,k$. The parameters $\boldsymbol{\alpha}$ are then found using the gradient descent method with update rule

$$
\alpha_{d,p} \leftarrow \alpha_{d,p} - \eta\frac{\partial\mathcal{C}}{\partial\alpha_{d,p}} \tag{38}
$$

where $\quad \eta = \beta\min(\Delta x)^{d-d_{\max}} \quad$ and $\quad \frac{\partial\mathcal{C}}{\partial\alpha_{d,p}} = (-1)^d\frac{1}{\Delta x\Delta t}\int f^p\frac{\partial^d\lambda}{\partial x^d}dxdt + 2\epsilon_0\alpha_{d,p} .$ (39)

This leads to the update rule for each coefficient, for example

$$
\begin{aligned}
\alpha_{1,1} &\leftarrow \alpha_{1,1} - \frac{\beta}{\min(\Delta x)^2}\frac{1}{\Delta x\Delta t}\int f\frac{\partial\lambda}{\partial x}dxdt - 2\beta\epsilon_0\alpha_{1,1} \\
\alpha_{1,2} &\leftarrow \alpha_{1,2} - \frac{\beta}{\min(\Delta x)^2}\frac{1}{\Delta x\Delta t}\int f^2\frac{\partial\lambda}{\partial x}dxdt - 2\beta\epsilon_0\alpha_{1,2} \\
\alpha_{1,3} &\leftarrow \alpha_{1,3} - \frac{\beta}{\min(\Delta x)^2}\frac{1}{\Delta x\Delta t}\int f^3\frac{\partial\lambda}{\partial x}dxdt - 2\beta\epsilon_0\alpha_{1,3} .
\end{aligned}
$$

### F.2 REACTION DIFFUSION SYSTEM OF EQUATIONS

As mentioned in Section B.4, for this case, we consider a system consisting of two PDEs, i.e. $N = \dim(\boldsymbol{f}) = \dim(\boldsymbol{p}) = 2$ where $\boldsymbol{f} = [f_1, f_2]$ and $\boldsymbol{p} = [p_1, p_2]$, in a two-dimensional input

space, i.e. $n = \dim(\boldsymbol{x}) = \dim(\boldsymbol{d}) = 2$ where $\boldsymbol{x} = [x_1, x_2]$, and $\boldsymbol{d} = [d_1, d_2]$. In addition, we consider candidate terms with derivatives such that $\boldsymbol{d} \in \mathcal{D}_{\boldsymbol{d}} = \{[0,0], [1,0], [0,1], [2,0], [0,2]\}$ and polynomials such that $\boldsymbol{p} \in \mathcal{D}_{\boldsymbol{p}} = \{[1,0], [0,1], [1,1], [2,0], [0,2], [2,1], [1,2], [3,0], [0,3]\}$. In other words, $d_{\max} = 2$ and $p_{\max} = 3$. The resulting forward model in Eq. 1 takes the form

$$\mathcal{L}_i[\boldsymbol{f}] = \partial_t f_i + \sum_{\boldsymbol{d}, \boldsymbol{p}} \alpha_{i, \boldsymbol{d}, \boldsymbol{p}} \nabla_{\boldsymbol{x}}^{(\boldsymbol{d})}[\boldsymbol{f}^{\boldsymbol{p}}] \tag{40}$$

where $i \in \{1, 2\}$, $\boldsymbol{f}^{\boldsymbol{p}} = f_1^{p_1} f_2^{p_2}$ and $\nabla_{\boldsymbol{x}}^{(\boldsymbol{d})} = \nabla_{x_1}^{(d_1)} \nabla_{x_2}^{(d_2)}$. This is equivalent to

$$\mathcal{L}_i[f_1, f_2] = \frac{\partial f_i}{\partial t} + \sum_{[d_1, d_2] \in \mathcal{D}_{\boldsymbol{d}}} \sum_{[p_1, p_2] \in \mathcal{D}_{\boldsymbol{p}}} \alpha_{i, [d_1, d_2], [p_1, p_2]} \frac{\partial^{d_1 + d_2} (f_1^{p_1} f_2^{p_2})}{\partial x_1^{d_1} \partial x_2^{d_2}}$$

$$= \frac{\partial f_i}{\partial t} + \alpha_{i, [0,0], [1,0]} f_1 + \alpha_{i, [0,0], [0,1]} f_2 + \alpha_{i, [0,0], [1,1]} f_1 f_2 + \ldots + \alpha_{i, [0,0], [0,3]} f_2^3$$

$$+ \alpha_{i, [1,0], [1,0]} \frac{\partial f_1}{\partial x_1} + \alpha_{i, [1,0], [0,1]} \frac{\partial f_2}{\partial x_1} + \alpha_{i, [1,0], [1,1]} \frac{\partial (f_1 f_2)}{\partial x_1} + \ldots + \alpha_{i, [1,0], [0,3]} \frac{\partial (f_2^3)}{\partial x_1}$$

$$+ \quad \ldots$$

$$+ \alpha_{i, [0,2], [1,0]} \frac{\partial^2 f_1}{\partial x_2^2} + \alpha_{i, [0,2], [0,1]} \frac{\partial^2 f_2}{\partial x_2^2} + \alpha_{i, [0,2], [1,1]} \frac{\partial^2 (f_1 f_2)}{\partial x_2^2} + \ldots + \alpha_{i, [0,2], [0,3]} \frac{\partial^2 (f_2^3)}{\partial x_2^2} \tag{41}$$

where $i \in \{1, 2\}$. As we can observe, we have $|\mathcal{D}_{\boldsymbol{d}}| \times |\mathcal{D}_{\boldsymbol{p}}| = 5 \times 9 = 45$ terms with unknown coefficients $\boldsymbol{\alpha}_i = [\alpha_{i, \boldsymbol{d}, \boldsymbol{p}}]_{d \in \mathcal{D}_{\boldsymbol{d}}, p \in \mathcal{D}_{\boldsymbol{p}}}$ for the $i$-th PDE, i.e. a total of 90 terms for the considered system, that we aim to find using the proposed adjoint method.

The cost functional in this case is simply

$$\mathcal{C} = \sum_{i=1}^{2} \left( \sum_{j,k} \left( f_i^*(\boldsymbol{x}^{(k)}, t^{(j+1)}) - f_i(\boldsymbol{x}^{(k)}, t^{(j+1)}) \right)^2 + \frac{1}{\Delta \boldsymbol{x} \Delta t} \int \lambda_i(\boldsymbol{x}, t) \mathcal{L}_i[\boldsymbol{f}(\boldsymbol{x}, t)] d\boldsymbol{x} dt \right) + \epsilon_0 \|\boldsymbol{\alpha}\|_2^2 . \tag{42}$$

The corresponding adjoint equation is given by

$$\frac{\partial \lambda_i}{\partial t} = \sum_{\boldsymbol{d}, \boldsymbol{p}} (-1)^{|\boldsymbol{d}|} \alpha_{i, \boldsymbol{d}, \boldsymbol{p}} \nabla_{f_i}[\boldsymbol{f}^{\boldsymbol{p}}] \nabla_{\boldsymbol{x}}^{(\boldsymbol{d})}[\lambda_i]$$

$$= \sum_{[d_1, d_2] \in \mathcal{D}_{\boldsymbol{d}}} \sum_{[p_1, p_2] \in \mathcal{D}_{\boldsymbol{p}}} (-1)^{d_1 + d_2} \alpha_{i, [d_1, d_2], [p_1, p_2]} \frac{\partial (f_1^{p_1} f_2^{p_2})}{\partial f_i} \frac{\partial^{d_1 + d_2} \lambda_i}{\partial x_1^{d_1} \partial x_2^{d_2}} \tag{43}$$

and $\lambda_i(\boldsymbol{x}^{(k)}, t^{(j+1)}) = 2(f_i^*(\boldsymbol{x}^{(k)}, t^{(j+1)}) - f_i(\boldsymbol{x}^{(k)}, t^{(j+1)}))$ for all $j, k$ and where $i \in \{1, 2\}$.

Assume, without loss of generality, that $i = 1$. Then, we can write

$$\frac{\partial \lambda_1}{\partial t} = + \alpha_{1, [0,0], [1,0]} \lambda_1 + \alpha_{1, [0,0], [1,1]} f_2 \lambda_1 + \alpha_{1, [0,0], [2,0]} (2f_1) \lambda_1 + \ldots + \alpha_{1, [0,0], [3,0]} (3f_1^2) \lambda_1$$

$$- \alpha_{1, [1,0], [1,0]} \frac{\partial \lambda_1}{\partial x_1} - \alpha_{1, [1,0], [1,1]} f_2 \frac{\partial \lambda_1}{\partial x_1} - \alpha_{1, [1,0], [2,0]} (2f_1) \frac{\partial \lambda_1}{\partial x_1} - \ldots - \alpha_{1, [1,0], [3,0]} (3f_1^2) \frac{\partial \lambda_1}{\partial x_1}$$

$$+ \quad \ldots$$

$$+ \alpha_{1, [0,2], [1,0]} \frac{\partial^2 \lambda_1}{\partial x_2^2} + \alpha_{1, [0,2], [1,1]} f_2 \frac{\partial^2 \lambda_1}{\partial x_2^2} + \alpha_{1, [0,2], [2,0]} (2f_1) \frac{\partial^2 \lambda_1}{\partial x_2^2} + \ldots + \alpha_{1, [0,2], [3,0]} (3f_1^2) \frac{\partial^2 \lambda_1}{\partial x_2^2} \tag{44}$$

and $\lambda_1(\boldsymbol{x}^{(k)}, t^{(j+1)}) = 2(f_1^*(\boldsymbol{x}^{(k)}, t^{(j+1)}) - f_1(\boldsymbol{x}^{(k)}, t^{(j+1)}))$ for all $j, k$. We can follow the same procedure for $i = 2$. The parameters $\boldsymbol{\alpha}_i$ are then found using the gradient descent method with update rule

$$\alpha_{i, \boldsymbol{d}, \boldsymbol{p}} \leftarrow \alpha_{i, \boldsymbol{d}, \boldsymbol{p}} - \eta \frac{\partial \mathcal{C}}{\partial \alpha_{i, \boldsymbol{d}, \boldsymbol{p}}} \tag{45}$$

where

$$\eta = \beta \min(\Delta \boldsymbol{x})^{|\boldsymbol{d}| - d_{\max}} \quad \text{and} \quad \frac{\partial \mathcal{C}}{\partial \alpha_{i,\boldsymbol{d},\boldsymbol{p}}} = (-1)^{|\boldsymbol{d}|} \frac{1}{\Delta \boldsymbol{x} \Delta t} \int \boldsymbol{f}^{\boldsymbol{p}} \, \nabla_{\boldsymbol{x}}^{(\boldsymbol{d})} [\lambda_i] d\boldsymbol{x} dt + 2\epsilon_0 \alpha_{i,\boldsymbol{d},\boldsymbol{p}} \quad (46)$$

with $\Delta \boldsymbol{x} = \Delta x_1 \Delta x_2$, leading to the update rule for each coefficient, for example

$$\alpha_{i,[0,0],[1,0]} \leftarrow \alpha_{i,[0,0],[1,0]} - \frac{\beta}{\min(\Delta \boldsymbol{x})^2} \frac{1}{\Delta \boldsymbol{x} \Delta t} \int f_1 \lambda_i d\boldsymbol{x} dt - 2\beta\epsilon_0 \alpha_{i,[0,0],[1,0]}$$

$$\alpha_{i,[1,0],[1,0]} \leftarrow \alpha_{i,[1,0],[1,0]} - \frac{\beta}{\min(\Delta \boldsymbol{x})} \frac{1}{\Delta \boldsymbol{x} \Delta t} \int f_1 \frac{\partial \lambda_i}{\partial x_1} d\boldsymbol{x} dt - 2\beta\epsilon_0 \alpha_{i,[1,0],[1,0]} \ .$$

