# OpenReview forum: "Data-Driven Discovery of PDEs via the Adjoint Method"
_ICLR.cc/2025/Conference — ICLR 2025 Conference Withdrawn Submission_

### Official Review · Reviewer_qLFP · 2024-10-29

**Soundness:** 3
**Presentation:** 3
**Contribution:** 2
**Rating:** 3
**Confidence:** 3

**Summary:**

An adjoint-based equation discovery framework is proposed in this paper to discover the underlying governing partial differential equations (PDEs) of physical systems. The proposed method initializes a general PDE by considering derivatives and polynomials of the derivatives up to a finite degree. A loss function is formed by incorporating the error between the prediction from the general PDE and the training labels and the corresponding adjoint equations. After that, a PDE-constrained optimization problem is solved to estimate the parameters of the initial PDE model, such that the solution of the general PDE will accurately approximate the training labels. The successful optimization yields coefficients of only the relevant derivatives to take significant values, thereby uncovering the underlying governing equation.

**Strengths:**

1. The idea of using adjoint equations to uncover underlying equations is interesting, and it seems to be new in the literature.
2. The theoretical derivations and analysis do a good job of explaining the key concepts of the framework.
3. Even in partial observations, when data at fine mesh is not available, the adjoint equations can discretize the general PDE on a finer mesh and, therefore, outperform other methods in the low data limit.

**Weaknesses:**

Please see below the comments that need to be further clarified.
1. The proposed framework is motivated by assuming a general PDE, which contains derivatives and their polynomials up to a certain degree. This is equivalent to basis functions in regression-based equation discovery frameworks, which is one of the major limitations of the basis-dependent discovery methods. Since, in an unknown scenario, the knowledge about the underlying physics will be minimal, one may need to consider a large number of basses/derivatives. In most of the examples, the authors consider only up to 3rd-order derivatives and their polynomials. Authors should consider a higher order of derivatives and polynomials to check the performance of their framework, both accuracy and computational efficiency.
2. The proposed framework closely resembles the PINN-SR algorithm. Therefore, a comparison with PINN-SR algorithms is necessary to gauge the effect of the adjoint equations.
3. The motivation for the equation discovery architecture is a little convoluted. The authors state that since data-driven simulators fail to learn the exact physics of dynamical systems, they fail to extrapolate beyond the training regime. Thus, we need an equation discovery framework, which post-discovery can be coupled with any standard numerical methods to obtain accurate predictions. While the second statement is true, it can be argued that after the discovery of a governing equation, one can still train a data-driven or physics-informed machine learning emulator for accelerating computational simulations. Therefore, the motivation needs to be reworked.
4. The Bayesian class of equation discovery algorithms also does a good job in distilling governing equation equations from data, particularly in noisy and low-data limits [1-3]. This paper should have discussed these frameworks. A comparison with a few such frameworks would further benefit the content of this paper.

[1] Nayek, Rajdip, et al. "On spike-and-slab priors for Bayesian equation discovery of nonlinear dynamical systems via sparse linear regression." Mechanical Systems and Signal Processing 161 (2021): 107986.\
[2] Zhang, Zhiming, and Yongming Liu. "Parsimony-enhanced sparse Bayesian learning for robust discovery of partial differential equations." Mechanical Systems and Signal Processing 171 (2022): 108833.\
[3] Hirsh, Seth M., David A. Barajas-Solano, and J. Nathan Kutz. "Sparsifying priors for Bayesian uncertainty quantification in model discovery." Royal Society Open Science 9.2 (2022): 211823.

**Questions:**

Please see below questions on the paper content:
1. line 87. PINN-SR uses the Adam optimizer, which can process data in batches. Thus, why do authors feel that PINN-SR will not scale well with the size of the data set?
2. line 135. The vector $\boldsymbol{f}^p$ may be missing a comma.
3. line 189. Does the framework work only for zero boundary conditions?
4. line 218. How $\sigma$ is selected. Consider an example of the Navier-Stokes equation, where the viscosity can be in the order of $10^{-4}$ to $10^{-5}$. In such cases, the proposed algorithm will discard the relevant terms.
5. line 282. Is this method applicable to time-independent systems like Darcy and Poisson's equations? If applicable, how averaging of gradients in algorithm 2 will be done in the absence of the time component.
6. Fig. 1(c). It is intriguing to see that the error increases with an increase in data in both methods.
7. Fig. 1(d). Since the proposed method uses a forward solver in the loop, is it not the computational time supposed to increase with increasing training samples?
8. Eq. (13-14). The authors show that even in ill-posed problems, the proposed method can provide an alternate sparse equation. However, estimating the error and visually observing the solution fields before concluding on the accuracy would be best.

**Limitations:**
The proposed framework does not show discovery from irregular observations (both in time and space). This should be included in the limitations.

---

> ### Author Response · Authors · 2024-11-15
> **Answer to qLFP**
>
> > 1. The proposed framework is motivated...
>
> **Answer:** Actually, we believe majority of physical processes are described with PDEs of 1st, 2nd, and rarely 3rd order derivatives. So, we do not agree with the referee. However, the adjoint method can be used for large class of PDEs, subject to the stability of the deployed numerical scheme. Please note that in appendix B.2, we have considered Kuramoto-Sivashinsky equation, which has derivatives up to 4th order, and leave more complicated test cases for future.
>
> > 2. The proposed framework closely resembles the PINN-SR algorithm....
>
> **Answer:** In the proposed method we are not learning any surrogate model. So we do not agree on the resemblance of proposed adjoint method with PINN-SR. However, we leave out such interesting comparisons to future work given the time constraint.
>
> > 3. The motivation for the equation discovery architecture is a little convoluted....
>
> **Answer:** We agree with the referee. We are trying to say that PDE discovery and data-driven surrogate models are answering two different questions. We added this to the introduction "Once the governing equation is found, one can either use the standard numerical methods for prediction, or train a PINN-like surrogate model for fast evaluation."
>
> > 4. The Bayesian class of equation discovery algorithms also does a good job...
>
> **Answer:** We thank the referee for pointing us to Bayesian class of equation discovery for PDE discovery. We added the citations in the introduction as related works. However, we leave further comparisons to future study given the time constraint.
>
> > line 87. PINN-SR uses the Adam optimizer, which can process data in batches. Thus, why do authors feel that PINN-SR will not scale well with the size of the data set?
>
> **Answer:** Although Adam optimizer eases the optimization problem by considering batches of data, the underlying cost function of the regression problem remains the same as PDE-FIND. Here, we make the claim and show that the explicit gradients computed from the adjoint method is more accurate that gradients computing from the point-wise L2 error using backpropagation.
>
> > line 135. The vector $f^p$  may be missing a comma.
>
> **Answer:** Thanks for spotting this typo. We corrected the manuscript.
>
> > line 189. Does the framework work only for zero boundary conditions?
>
> **Answer:** The compact support of $\lambda$ is motivated by the fact that we consider boundary conditions as known. In case boundary conditions are also parameterized, we need to add another constraint with its Lagrange multiplier to find its parameters. We added this justification to the text.
>
> > line 218. How $\sigma$ is selected. Consider an example of the Navier-Stokes equation, where the viscosity can be in the order of $10^{-4}$ to $10^{-5}$. In such cases, the proposed algorithm will discard the relevant terms.
>
> **Answer:** As mentioned in the text, $\sigma$ is a user-defined threshold, to drop out terms that have close to zero coefficients in return. In several cases, we showed that adjoint method finds coefficients of unimportant terms to be less than $10^{-8}$, see results for Burgers' and heat equation. This would allow finding PDEs with such small coefficieint.
>
> > line 282. Is this method applicable to time-independent systems like Darcy and Poisson's equations? If applicable, how averaging of gradients in algorithm 2 will be done in the absence of the time component.
>
> **Answer:** We thank the referee for their question. While we restricted ourself to temporal-spatial PDEs, one may consider time-independent problems by introducing artificial time, such that at the steady state, the time-independent systems is recovered. This is definitely an interesting problem, and we may investigate further in the future work.
>
> > Fig. 1(c). It is intriguing to see that the error increases with an increase in data in both methods.
>
> **Answer:**  The increase of error versus data size for PDE-FIND method is clearly visible. However. for the adjoint method, the error stays on the same order of magnitude.
>
> > Fig. 1(d). Since the proposed method uses a forward solver in the loop, is it not the computational time supposed to increase with increasing training samples?
>
> **Answer:** Yes, the computational time increases for both PDE-FIND and adjoint method in Fig1.d.
>
> > Eq. (13-14). The authors show that even in ill-posed problems, the proposed method can provide an alternate sparse equation. However, estimating the error and visually observing the solution fields before concluding on the accuracy would be best.
>
> **Answer:** In the case of the considered ill-posed problem, since we know the solution f, the error at any point can be obtained by plugging in $f=sin(x-t)$ into found PDE. For example, in case of 13 and 14, $|\cos(x-t)(-1+0.996)| \leq |\cos(x-t)(-1+0.989)|$ at all x and t.

---

> ### Comment · Reviewer_qLFP · 2024-11-20
> **Reply to the author responses:**
>
> Thanks for the point-wise reply! However, I am still looking for a better reply.
>
> + "Actually, we believe majority of physical processes are described with PDEs of 1st, 2nd, and rarely 3rd order derivatives..."
>   - **Comment**: The illustration of the Kuramoto-Sivashinsky equation is very well appreciated. However, observe that the order and degree of derivatives are chosen so that the number of basis functions is limited to 8 to 9. Besides the order, the interaction between the derivatives and polynomials is also common. Examples are the Navier-Stokes equation, Phase-field modeling, Modelling of Instability in boundary layer flow, etc. The problem is that when such terms are included, the correlations between these terms hinder the identification of the true governing equation. Identification of wrong terms may result in overfitting in the long prediction horizon. Thus, it becomes necessary to determine whether the adjoint method can tackle these problems of existing equation discovery methods. Another reason behind including higher-order terms is to increase the capability of the proposed method. While we know that mathematical PDE models are full of assumptions, we should expect the proposed framework to provide us with a more accurate model in the unknown real-world scenarios.
>
> + "In the proposed method we are not learning any surrogate model..."
>   - **Comment**: It is mentioned in the PINN-SR paper that PINN-SR aims to simultaneously model the system response and identify the parsimonious closed form of the governing PDE(s). Although the response modeling part is irrelevant in the context of the proposed work, the identification of parsimonious forms using basis functions is quite similar to the proposed work. Therefore, saying that there is no resemblance does not properly justify the rebuttal, in my opinion.
>
> + "We thank the referee for pointing us to Bayesian class of equation discovery for PDE discovery..."
>   - **Comment**: Having highlighted the time constraint, at least one comparison with one of the Bayesian methods is necessary to understand the need for such algorithms.
>
> + "Although Adam optimizer eases the optimization problem by considering batches of data..."
>   - **Comment**: Do the authors mean that the loss function in PINN-SR is estimated over the entire dataset in a single batch? Do the authors verify it in the PINN-SR algorithm, including the released source codes? Secondly, how the authors come to the conclusion that the adjoint gradients are more accurate than point-wise L2 error using backpropagation. Maybe I missed it in the paper.
>
> + "Yes, the computational time increases for both PDE-FIND and adjoint method in Fig1.d."
>   - **Comment**: The authors agree that the computational time is supposed to increase with increasing training samples. However, it is not the case in all the examples. For e.g., see Fig. 7 (Burgers' equation). Can authors comment on this?
>
> + "In the case of the considered ill-posed problem, since we know the solution f,..."
>   - **Comment**: In the case of the considered ill-posed problem, I agree the error, although marginal, in the identified expression from the proposed adjoint method is less than that of the PDE-FIND method. However, the question is whether the identified reduced-order equation can actually represent the true system. The authors should compare the responses of the identified and true systems. This way, it can examined whether the proposed adjoint method can handle ill-posed problems. Otherwise, the claim should be removed from the paper.

---

> ### Author Response · Authors · 2024-11-23
>
> > However, observe that the order and degree of derivatives are chosen so that the number of basis functions is limited to 8 to 9. ...
>
> Please note that in Appendix B.4 for the Reaction Diffusion system of equations, we showed a case of complicated PDE where the adjoint method can find the correct PDE  given a guessed parameterized PDE with 90 terms with unknown coefficients that include cross all the derivatives up to 2nd order and polynomial up to 3rd order.
>
> > Examples are the Navier-Stokes equation, Phase-field modeling, Modelling of Instability in boundary layer flow, etc....
>
> I guess I misunderstood the referee. Does the Navier Stokes equations have terms with derivatives higher than 2nd order? In either case, it is possible to include higher order terms in the guessed PDE. The only draw back is that to guarantee stability of the finite difference method, one has to choose a small time step size which increases the computational cost. We think we have shown enough complicated examples as a proof-of-concept in the manuscript.
>
> >  It is mentioned in the PINN-SR paper that PINN-SR aims to simultaneously model the system response and identify the parsimonious closed form of the governing PDE(s). Although the response modeling part is irrelevant in the context of the proposed work, the identification of parsimonious forms using basis functions is quite similar to the proposed work. Therefore, saying that there is no resemblance does not properly justify the rebuttal, in my opinion.
>
> > Having highlighted the time constraint, at least one comparison with one of the Bayesian methods is necessary to understand the need for such algorithms.
>
> We thank the referee for their comment. Although it would be interesting to see how the adjoint method would compare to other methods such as PINN-SR or Bayesian methods, we leave such comparisons to future work. For the reader, the comparisons with the state-of-art and benchmark PDE-FIND should be enough as a point of reference.
>
> > Do the authors mean that the loss function in PINN-SR is estimated over the entire dataset in a single batch? Do the authors verify it in the PINN-SR algorithm, including the released source codes? Secondly, how the authors come to the conclusion that the adjoint gradients are more accurate than point-wise L2 error using backpropagation. Maybe I missed it in the paper.
>
> Our understanding is that PDE-FIND as well as PINN-SR try to find the fit that minimizes L2 error on all points by directly solving the optimization problem. Using mini-batches or not, i.e. using gradient descent or stochastic gradient descent, finding the global minimum becomes difficult as number of unknown parameters and data points increases. We believe this is a common and known problem with optimization problem. In the case of PINN-SR, the total loss function is formulated as (eq. 3 in [Nature communications, 12(1), 6136.])
>
> $L_d + \alpha L_p + \beta ||\Lambda||_0$
>
> where $L_d $ is the L2 error between model prediction and data, $L_p$ is a PINN loss for the intermediate found PDE, $\alpha$ is the relative weighting of the residual physics loss function, and  $\beta ||\Lambda||_0$ is the regularization term.
> Given the data set, the total loss is computed considering the same $\alpha$ as a hyper-parameter and constant for all points.
>
> In the adjoint method, we take $\alpha$ as the Lagrange multiplier (denoted in our paper by $\lambda$ as function of (x,t)), and find a relation between its value at each (x,t) and the point-wise error. This is achieved by taking variational derivatives, finding the adjoint relations (eq. 7 and 8), and solving the backward equation. By doing so, we find out how the point-wise error propagates forward/backwards in time. This allows us to find the optimal gradient of the total cost function with respect to the parameters.
>
> > The authors agree that the computational time is supposed to increase with increasing training samples. However, it is not the case in all the examples. For e.g., see Fig. 7 (Burgers' equation). Can authors comment on this?
>
> By increasing the resolution of the spatial mesh, the numerical error in computing the gradients decreases (eq. 4). Also, by considering more time steps, in each epoch the parameters $\alpha$ are updated more often, see Algorithm 1. The combination of these two allows adjoint method to converge in less number of epochs as the data on a finer spatial-temporal mesh is available, although the cost of each epoch is increased. In our experiment, the overall cost seems to be independent of the number of discrete points in time. However, the cost seems to increase when the spatial grid gets finer. See Fig.1(d), Fig. 2(d), Fig. 7(right) and note the number of discrete points in t and x denoted by $(N_t, N_x)$. Overall, we see better performance than PDE-FIND as the size of dataset increases.

---

> > ### Author Response · Authors · 2024-11-23
> >
> > > The authors should compare the responses of the identified and true systems....
> >
> > Do you mean plotting the function values of the discovered PDE against the data? I am not sure if that adds value to the paper, since we can quantify the error. Again, since we know the exact solution to the PDE, the residual error in any norm can be obtained by plugging the true solution in the discovered PDE for either method. Can the referee clarify further?

---

> > > ### Comment · Reviewer_qLFP · 2024-11-25
> > > **Reply to the author responses:**
> > >
> > > If I have understood correctly, Eq. (13) tries to approximate Eq. (10). Now observe that the Heat equation in section 3.1 has a second-order derivative, whereas the equation identified by the proposed Adjoint and PDE-FIND has only a first-order derivative. This raises the concern of whether the first-order derivative can account for the effect of the original Heat equation. Therefore, a direct comparison with ground truth solutions, whether in terms of plots or errors, will provide a better understanding.

---

> > > > ### Author Response · Authors · 2024-11-25
> > > >
> > > > In section 6, we look at wave equation as an example and address the issue with ill-posedness of finding PDE given data. Eq. 13 and 14 are discovered PDEs given the discrete points of sin(x-t) which is the exact solution to a family of PDEs (eq.12) that include wave equation $f_t+f_x=0$ among other. Here, we show that both adjoint and PDE-FIND find the PDE with the least number of terms (wave equation) in the family of permutable PDEs thanks to the regularization term. Please note that Heat equation (eq.10) does not permit solution of the form $sin(x-t)$.

---

### Official Review · Reviewer_91Pr · 2024-11-02

**Soundness:** 2
**Presentation:** 1
**Contribution:** 2
**Rating:** 3
**Confidence:** 5

**Summary:**

This paper considers the problem of identifying the parameters of a system of PDEs based on measurement data. The proposed method fits the solution of the PDE to the data while adapting its parameters. It uses the adjoint method that combines a data goodness-of-fit objective with a PDE satisfaction constraint using a Lagrangian formulation. Using variational techniques, it is then possible to obtain evolution equations for those weights (multipliers), which then allow the gradient objective to be computed with respect to the PDE parameters. The proposed method is compared with PDE-FIND for a number of PDEs in the standard setting as well as when the temporal data is coarse and when the data is noisy.

**Strengths:**

- The proposed method archives good results on the considered PDEs considered compared to PDE-FIND
- The experiments consider relevant settings, including coarse temporal resolution, noisy data, and ill-posed tasks
- Using the adjoint method, it is possible find straightforward algorithms based on gradient descent

**Weaknesses:**

1. The algorithms proposed in the work implicitly (i) solve the forward PDE (1), which can be challenging and time consuming (line 242); (ii) solve the adjoint PDE model (7), which again can be challenging and time consuming (line 243); and (iii) compute gradients of the adjoint variables $\lambda$, which are only know implicitly through (7) (line 244). The manuscript does not thoroughly discuss how this affects the computational complexity of the proposed method and whether the choice of solution algorithm could affect the performance of the proposed method. This is particularly important since other methods, such as PDE-FIND, do not appear to require such explicit forward evolutions.
It is not clear from the manuscript how the proposed method differs from classical adjoint methods, proposed originally in the context of design or shape optimization (see, e.g., Jameson 2003). The text only mentions that “unlike the usual use of PDE-constrained adjoint optimization where the governing equation is known, in this paper we are interested in finding the form along with the coefficients of the PDE given data.” Nevertheless, the parameters of the PDE can be mapped directly to design parameters used in traditional adjoint methods. As such, there doesn’t seem to be any challenge requiring substantial contributions in the setting of this paper.

1. The paper is not clearly written. There are several typos (“Eq. equation” in lines 201, 215, 376, 482, 1172) and odd uses of language (“Here, $x(k)$ is coordinates”; ) that make it hard to follow. Many elements are either undefined or defined implicitly/informally. For instance, in (2) are those gradient products or an iterated differentiation (as in multi-index differentials from Sobolev spaces)? What is meant by “semi-discrete total variation of C” (line 176)? The definition is given without a reference or explanation. Additionally, the structure of the experiments is not clear as Section 3 named “Results” is followed by Section 4,5,6 named “Partial observations in time”... all containing experimental results as well. This lack of care in the writing renders the paper confusing for the reader.

1. Many results are provided without derivations or references. Since their definitions are incomplete, it is impossible to judge their validity. For instance, the derivations of (4) and (6) are only informally explained. And since nowhere is it defined in which functional space in which the solution $f$ of (1) must be in, it is impossible to judge the correctness of (4), even if the reader had to assume the smoothness/differentiability of $\lambda$. Additionally, since conditions such as $\lambda(x,t) \to 0$ for $x \in \partial\Omega$ are not justified and without knowing the functional (metric) space in which $\lambda$ is optimized, it is impossible to guarantee its differentiability in (4) and (6). Similar comments apply to the derivation of (7)-(8) from (6).

1. The manuscript only considers very simple PDEs. In particular, the main text only tackles the 1D heat equation with D=1, the 1D Burgers equation, and a simple 1D wave equation. These are low-dimensional, simple first-order equations with very few parameters to fit. In fact, the proposed method does not appear to be able to handle PDEs beyond linear, first order in time (particularly since it must solve (1) explicitly). In particular, it does not consider very overparametrized models either (large $p$ or $d$). Hence, the performance of the method in PDEs whose solutions are challenging or where there is substantial uncertainty on the model (large $p$ or $d$) is not tested.

1. Point (4) is particularly critical given that the proposed method is only compared against one baseline (PDE-FIND) which is already somewhat dated (2017). Many other baselines are available, in particular, those detailed in the introduction. It is unclear why none were used in the experiments.

1. There is only very limited discussion on the use of l2 regularization. Particularly since it is well-known from polynomial fitting that the l2-norm is not well-adapted to find sparse solutions, i.e., where many values are 0. That is why, e.g., Brunton et al. 2016 use the l1 norm. While the thresholding heuristic employed by the paper is reminiscent of iterative hard thresholding (IHT) methods from compressed sensing (see, e.g., Blumensath and Davis, Applied Computational Harmonic Analysis, 2009) it does not inherit the same guarantees or theoretical advantages. In fact, this relation is not even mentioned in the manuscript.

**Questions:**

See Weakness above. We include here some additional minor issues:

1. When considering sensitivity to noise (line 411), the noise variance is specified in percentage. With respect to what? Is it an SNR?
1. Where does the normalization of the integral in (3) come from? This doesn’t seem to make sense given that the manuscript considers a regular grid.
1. When discussing the learning rate, it is mentioned that “the gradient of the cost function is most sensitive to the highest order terms of the PDE.” Why is that the case?
1. When defining $f^*$ on line 121, shouldn’t it be $f^*: \Omega \times [0,T] \to \mathbb{R}^N$? As it is, the desired solution is only on $\mathcal{G}$. How would that be consistent with, e.g., coarser time grids?

---

> ### Author Response · Authors · 2024-11-15
>
> > 1. The algorithms proposed in the ...
>
> **Answer:** In line 227-234, we discuss the numerical complications in the proposed adjoint method for PDE discovery task. We note that although each iteration of adjoint method is more expensive compared to PDE-FIND, in several test cases we showed that the method converges relatively fast (in less than 100 iterations/epochs).
>
> As mentioned in contributions, the adjoint method has not been used in the literature to discover PDEs from data, and that is the novelty of this paper. If the referee disagrees, we kindly ask them to point us to a paper that uses adjoint method for PDE discovery task.
>
> > The paper is ...
>
> **Answer:** Thanks for spotting the typos. We have corrected them in the revised version.
> In **Problem setup.** we explained the deployed notation. That should clarify referees confusion.
>
> Eq.(2) denotes iterated differential operator. We added this to the text.
>
> Please note that eq.(3) is semi-discrete, i.e. the first term includes $L^2$ error between solution of PDE and data points on discrete points (with superscript k for spatial and j for temporal indices), and the second term (constraint on PDE) is continuous. We think the equation illustrate that clearly.
>
> We thank the referee for their suggestion about arrangement of sections. We changed section 4,5,6 into subsections of section 3.
>
> > 3. Many results ...
>
> **Answer:** Eq. (4) is obtained by taking derivative of Eq. (3) with respect to parameters $\alpha$, and shifting derivatives from $f$ to $\lambda$ using integration by parts and enforcing $\lambda\rightarrow 0$ for $x\in \partial \Omega$. Clearly, we are assuming that all considered derivatives of $\lambda$ exist, and $\lambda\rightarrow 0$ on $ \partial \Omega$. That defines the functional space of $\lambda$. Similar steps, integration by parts and letting  $\lambda\rightarrow 0$ for $x\in \partial \Omega$, are considered to obtain eq. 6, besides functional derivative of $f$. We believe these derivations are trivial and leave it to the reader.
>
> The compact support of $\lambda$ is motivated by the fact that we consider boundary conditions as known. In case boundary conditions are also parameterized, we need to add another constraint with its Lagrange multiplier to find its parameters. We added this justification to the text.
>
> > 4. The manuscript only considers very simple PDEs...
>
> **Answer:** We agree with the referee that more tests on more complex dynamics can improve the paper. We kindly remind the referee that in Appendix B.4, we actually have shown an example of 2-dimensional coupled system of PDEs, i.e. reaction diffusion system of equations. Please see other challenging problems in Appendix B. We leave more complicated test cases for future work given the time constraint.
>
> > 5. Point (4) is particularly critical...
>
> **Answer:** To the best of our knowledge, PDE-FIND is the state-of-the-art method for PDE reconstruction, and that is why we only compared our method to that as a point of reference.
>
> > There is only very limited discussion on the use of l2 regularization...
>
> We thank the referee for their comment. We note that if the data can be represented with a unique PDE, then the adjoint method does not need the regularization factor at all. In fact, in Appendix D, Fig 18, we showed that by letting $l^2$ regularization factor to zero, adjoint method is capable of finding the underlying PDE as a solution to the sparse search. We only introduced $l^2$ regularization to distinguish between PDEs in the pathological cases, where the solution is not unique. Similarly, the thresholding is only used to make the sparse search faster, dropping out irrelevant terms. Without it, the adjoint method still works, but may converge in more interations. We thank the referee for pointing us to IHT method. We have added the reference to the manuscript.
>
> >When considering sensitivity to noise (line 411), the noise variance is specified in percentage. With respect to what? Is it an SNR?
>
> We introduce noise $\epsilon\sim \mathcal{N}(0,\sigma^2)$ to each data point $f^*$ by $f^*(1+\epsilon)$. Therefore, the added term $\epsilon f^*$ to the signal $f^*$ has the noise/signal ratio of $\epsilon$.
>
> > Where does the normalization of the integral in (3) come from? This doesn’t seem to make sense given that the manuscript considers a regular grid.
>
> **Answer:** We agree with the referee. We removed the normalization factor in Eq. 3. Given we consider normalized settings, our results do not change.
>
> > When discussing the learning rate...
>
> **Answer:**  This was an observation we made in this work. We clarified that in the revised version.
>
> > When defining $f^*$ on line 121,...
>
> **Answer:** Actually, $f^*$ are discrete points defined on the grid. The coarser time grids would work because $f^*$ only appears as the final condition for the adjoint equation, and is not present in neither the forward nor backward equations.

---

> > ### Comment · Reviewer_91Pr · 2024-11-25
> >
> > > In line 227-234, we discuss the numerical complications ...
> >
> > **Reviewer:** I thank the authors for the clarification and their comment in line 227-234 is appreciate. This is however, not a sufficiently broad analysis of the computational complexity of the method. A direct comparison to PDE-FIND (at least) should be provided, be it in terms of complexity or run time. The fact that the adjoint method can provide more accurate estimates than, e.g., PDE-FIND, at a (potentially considerably) higher computational cost raises a question of compromise that is fundamental. It is hard to judge the value and use cases of these methods if they are not compared on equal grounds. I believe this would require a major revision of the manuscript.
> >
> >
> > > As mentioned in contributions, the adjoint method has not been used in the literature to discover PDEs from data, and that is the novelty of this paper. If the referee disagrees, we kindly ask them to point us to a paper that uses adjoint method for PDE discovery task.
> >
> > **Reviewer:** The adjoint method is a general PDE-constrained optimization method used to solve inverse problems. My point was simply that inverse problems are absolutely equivalent to "parameter identification." To be specific, consider the application of the adjoint method to photonics in
> >
> > [Hughes et al., "Adjoint method and inverse design for nonlinear nanophotonic devices," 2018]
> >
> > The $\phi$ in (1) are arbitrary and could represent parameters of the underlying PDE. In fact, they do since they will permittivity coefficients. As such, their derivations in Section 2 are essentially the same as those in Section 2 of the current manuscript. Even their loss function in (20) is the same quadratic as the one in (3) in the current manuscript (modulo the weights $m$).
> >
> > My point is not that [Hughes et al., 2018] is a particularly relevant reference (it was found by a search on google scholar). But it is fully representative of the adjoint method bibliography, which is why I asked whether there were new specific challenges requiring substantial technical contributions in this paper. I may have missed something, but the derivations in Section 2 appear to be quite standard (see, e.g., Chapter 8 in [Strang, "Computational Science and Engineering"])
> >
> >
> > > Eq. (4) is obtained by taking derivative of Eq. (3) ...
> >
> > **Reviewer:** I understand the manipulations as they are quite typical from weak formulations. I therefore believe that all the steps taken by the authors are *justifiable*. That is why I believe this is a minor issue. Still, they need to be *justified*. As it is, there are a lot of informal statements in the manuscript. I understand this may seem an overly pedantic point, but "clearly, we are assuming that all considered derivatives of $\lambda$ exist" is not a rigorous statement. Particularly since an overly smooth $\lambda$ imposes stronger restrictions the space of (weak) solutions considered. I encourage the authors to make their claims rigorous at the very least in the appendices.

---

> > > ### Author Response · Authors · 2024-11-26
> > >
> > > We thank the referee for their constructive comments.
> > >
> > > It is not so trivial to compare the cost for the same error between the two method. However, throughout the paper we compared error and execution time for converged solution of both adjoint method and PDE-FIND. For comparisons of execution time and error for the considered test cases, please see Fig1c-d, Fig2c-d, Fig4c-d, Fig.7, Fig.9, Fig.11 and Fig.16.
> > >
> > > We agree with your comments, and will revise the paper to address your concerns on the complexity analysis, detailed derivation of adjoint equation, and specifying the functional spaces for the next submission.

---

### Official Review · Reviewer_x4HQ · 2024-11-04

**Soundness:** 2
**Presentation:** 2
**Contribution:** 2
**Rating:** 3
**Confidence:** 5

**Summary:**

This paper presents a method for reconstructing a parameterized partial differential equation (PDE) from observed data using the adjoint method. The authors claim that their approach offers a new perspective on PDE reconstruction by leveraging adjoint-based optimization, demonstrating the method on simple examples and show better performance compared to the well known PDE-FIND.

**Strengths:**

The paper is generally well-written, with clear explanations of the methodology and a solid foundation in the adjoint approach. The derivations are technically sound and should be accessible to readers familiar with inverse problems and PDEs.

**Weaknesses:**

(1) The approach presented here is relatively incremental, as the reconstruction of parameterized PDEs has been extensively studied within the inverse problem community for several decades. Many of the concepts explored, particularly adjoint-based parameter estimation, are already well-established. The paper would benefit from a more explicit discussion of how this work advances or differs from existing methods in the literature on PDE-constrained optimization and inverse problems.
(2) The demonstration example is limited to simple lower dimensional problems, which may not be sufficient to convincingly illustrate the method’s robustness or scalability to higher-dimensional, real-world PDEs. Given the computational efficiency implied by the adjoint method, testing on a more challenging, higher-dimensional example or a PDE with more complex dynamics would strengthen the paper. This would also allow for a more thorough evaluation of the method's effectiveness in a broader range of realistic scenarios.
(3) The paper would benefit from a comparative analysis with other state-of-the-art methods for PDE reconstruction. This comparison could help clarify the unique contributions and limitations of the adjoint-based approach relative to current methods in data-driven PDE reconstruction.

**Questions:**

(1) Can you provide more complex examples in high dimensional space?
(2) Can you provide some a posteriori analysis analysis regarding the learning performance with respect to the given data or type of PDEs?
(3) Can you derive some well posed-ness of the parametrization of PDE? i.e, if I provide two similar parameterized PDEs, how about the learning performances regarding interpolation and prediction?

---

> ### Author Response · Authors · 2024-11-15
> **Answer to x4HQ**
>
> > (1) The approach presented here is relatively incremental, as the reconstruction of parameterized PDEs has been extensively studied within the inverse problem community for several decades. Many of the concepts explored, particularly adjoint-based parameter estimation, are already well-established. The paper would benefit from a more explicit discussion of how this work advances or differs from existing methods in the literature on PDE-constrained optimization and inverse problems.
>
> **Answer:** As mentioned in **contributions**, the adjoint method has not been used in the literature to discover PDEs from data, and that is the novelty of this paper. If the referee disagrees, we kindly ask them to point us to a paper that uses adjoint method for PDE discovery task.
>
> > (2) The demonstration example is limited to simple lower dimensional problems, which may not be sufficient to convincingly illustrate the method’s robustness or scalability to higher-dimensional, real-world PDEs. Given the computational efficiency implied by the adjoint method, testing on a more challenging, higher-dimensional example or a PDE with more complex dynamics would strengthen the paper. This would also allow for a more thorough evaluation of the method's effectiveness in a broader range of realistic scenarios.
>
> **Answer:** We agree with the referee that more tests on more complex dynamics can improve the paper. We kindly remind the referee that in section B.4, we actually have shown an example of 2-dimensional coupled system of PDEs, i.e. reaction diffusion system of equations. We leave more complicated test cases for future work given the time constraint.
>
> > (3) The paper would benefit from a comparative analysis with other state-of-the-art methods for PDE reconstruction. This comparison could help clarify the unique contributions and limitations of the adjoint-based approach relative to current methods in data-driven PDE reconstruction.
>
> **Answer:** To the best of our knowledge, PDE-FIND is the state-of-the-art method for PDE reconstruction. We would like to ask the referee to be more specific about their request. Which other method should we consider for comparison besides PDE-FIND?
>
>
> > (1) Can you provide more complex examples in high dimensional space?
>
> **Answer:** In section B.4, we actually have shown an example of 2-dimensional coupled system of PDEs, i.e. reaction diffusion system of equations.
>
> > (2) Can you provide some a posteriori analysis analysis regarding the learning performance with respect to the given data or type of PDEs?
>
> **Answer:** We are not sure what the referee means. The performance of adjoint method is affected by the choice of the numerical methods, which depends on the type of PDE. However, we showed that a simple finite difference method is good enough in recovering underlying PDEs, as long as we pick small enough $\Delta t$ to ensure stability.
>
> > (3) Can you derive some well posed-ness of the parametrization of PDE? i.e, if I provide two similar parameterized PDEs, how about the learning performances regarding interpolation and prediction?
>
> **Answer:** We would like to ask the referee to clarify what they mean. In the current manuscript, we have provided an analysis on regularizing ill-posed PDE discovery task in section 6, and incomplete guessed PDE space in Appendix C. Furthermore, in section 4, we addressed the case when we are given partial observations in time. That may answer their question about performance versus interpolation.

---

> > ### Comment · Reviewer_x4HQ · 2024-11-22
> > **Further question: well posedness of the the parameterized PDE**
> >
> > Basically, I would like to see the impact of
> > (1) imperfect learning when fixed basis but getting two different sets of parameters when using different optimization algorithm or learning rate, etc.
> > (2) parametrize the PDE in two different ways, also a follow-up question regarding any a posteriori analysis regarding the choice of parametrization basis.

---

> > > ### Author Response · Authors · 2024-11-23
> > >
> > > > Basically, I would like to see the impact of (1) imperfect learning when fixed basis but getting two different sets of parameters when using different optimization algorithm or learning rate, etc.
> > >
> > > In Appendix D, we have analyzed the error of proposed adjoint method against different hyper parameters, such as regularization factor, threshold tolerance (Fig. 18), and learning rate (Fig. 19) for Burgers' and Kuramoto Sivashinsky equation. We believe these results should clarify some of the doubts.
> > >
> > > > (2) parametrize the PDE in two different ways, also a follow-up question regarding any a posteriori analysis regarding the choice of parametrization basis.
> > >
> > > In Appendix C, we took the data from Burgers' equation and considered two parameterized PDEs, one including df^2/dx term and the other excluding this term. We called the latter incomplete guessed PDE space as the exact term is missing in the guessed PDE form. While the adjoint method can find the exact solution given the complete guessed PDE form, the latter converges to another PDE that has relative error of O(10^-5) from data. The fact that the point-wise error during training does not decrease any further (right figures in Fig.17) is an indication of either the considered PDE is missing a term or obtained error is of the same order as the noise in data set (which is not the case here). We believe this result should answer the referee's doubt about having two different parameterization of PDE.

---

> ### Comment · Reviewer_x4HQ · 2024-11-23
>
> I would to see some error analysis to support the choice of basis here. Also, I feel that pointwise error is not very convincing from classical numerical PDE perspective. I feel that the design and learning could be benefit more from numerical PDE. And some evidence should be displayed to support machine learning is better than classical numerical PDE methods.

---

> > ### Author Response · Authors · 2024-11-26
> >
> > We thank the referee for raising their concern. Although, we are not sure how to address
> >
> > > And some evidence should be displayed to support machine learning is better than classical numerical PDE methods.
> >
> > we will revise the manuscript and add more error analysis for our next submission.

---

### Official Review · Reviewer_2rUM · 2024-11-05

**Soundness:** 2
**Presentation:** 2
**Contribution:** 2
**Rating:** 3
**Confidence:** 4

**Summary:**

The authors claim that they can discover some PDE via an adjoint-based method.

**Strengths:**

The topic looks very interesting.

**Weaknesses:**

The manuscript has not been well-written, so that the reader cannot find their motivation clearly.
The theoretical part has been shown rigorously.
The experiment part: description not clear

**Questions:**

Hopefully, the authors can make your contribution clearly and make your experiment and theories to test your results.

---

> ### Author Response · Authors · 2024-11-15
> **Answer to 2rUM**
>
> > The manuscript has not been well-written, so that the reader cannot find their motivation clearly. The theoretical part has been shown rigorously. The experiment part: description not clear
>
> > Hopefully, the authors can make your contribution clearly and make your experiment and theories to test your results.
>
> We thank the referee for their comment. However, we disagree with them. We believe the test cases are actually detailed, please see Appendix B. Can the referee be more specific? What details are missing?

---

### Note · Authors · 2024-11-26

**Comment:**

We have decided to withdraw the paper and revise the manuscript, and submit it to another venue. We thank the referees for their constructive comments that helped us with improving our paper.

Having said that, we would like to bring it to the editor's attention that we felt some bias from a reviewer (x4HQ) with comments like

"And some evidence should be displayed to support machine learning is better than classical numerical PDE methods."

or the other reviewer (2rUM) saying

"The manuscript has not been well-written, so that the reader cannot find their motivation clearly. The theoretical part has been shown rigorously. The experiment part: description not clear"
"Hopefully, the authors can make your contribution clearly and make your experiment and theories to test your results."

without engaging in a conversation.

We believe regardless of how a method is called, we should be fair and acknowledge strengths and weaknesses as we tried to do so in this manuscript. However, if a classical method outperforms a known machine learning method for a specific task in some limit, there is no shame in admitting that.

**Withdrawal Confirmation:**

I have read and agree with the venue's withdrawal policy on behalf of myself and my co-authors.